# A subgroup of light-driven sodium pumps with an additional Schiff base counterion

E. Podoliak [1,14], G. H. U. Lamm [2,14], E. Marin[3], A. V. Schellbach[2,13], D. A. Fedotov [4], A. Stetsenko [3], M. Asido [2], N. Maliar[5], G. Bourenkov[6], T. Balandin [7,8], C. Baeken[7,8], R. Astashkin[9], T. R. Schneider [6], A. Bateman [10], J. Wachtveitl [2], I. Schapiro [4], V. Busskamp [1], A. Guskov [3], V. Gordeliy[7,8,9], A. Alekseev [11,12] ✉ & K. Kovalev [6] ✉

Light-driven sodium pumps (NaRs) are unique ion-transporting microbial rhodopsins. The major group of NaRs is characterized by an NDQ motif and has two aspartic acid residues in the central region essential for sodium transport. Here we identify a subgroup of the NDQ rhodopsins bearing an additional glutamic acid residue in the close vicinity to the retinal Schiff base. We thoroughly characterize a member of this subgroup, namely the protein *Er*NaR from *Erythrobacter sp. HL-111* and show that the additional glutamic acid results in almost complete loss of pH sensitivity for sodium-pumping activity, which is in contrast to previously studied NaRs. *Er*NaR is capable of transporting sodium efficiently even at acidic pH levels. X-ray crystallography and single particle cryo-electron microscopy reveal that the additional glutamic acid residue mediates the connection between the other two Schiff base counterions and strongly interacts with the aspartic acid of the characteristic NDQ motif. Hence, it reduces its pKa. Our findings shed light on a subgroup of NaRs and might serve as a basis for their rational optimization for optogenetics.

Light-driven sodium pumps (NaRs) were discovered in 2013 with the characterization of the microbial rhodopsin (MR) KR2 from the bacterium *Krokinobacter eikastus*[1]. NaRs are membrane proteins that actively transport sodium outside of the cell in response to light illumination[2]. Since 2013, numerous various NaRs have been identified[3–6]. Most of them have an NDQ motif (N112, D116, and Q123 residues in KR2 corresponding to D85, T89, and D96 residues of the proton pump bacteriorhodopsin (BR[7])) in the helix C. Another group of MRs able to pump sodium was recently reported to have a DTG motif[8].

As all other MRs, the structure of NDQ rhodopsins consists of seven transmembrane helices (A-G) encapsulating a retinal cofactor,

[1]Department of Ophthalmology, University Hospital Bonn, Medical Faculty, Bonn, Germany. [2]Institute of Physical and Theoretical Chemistry, Goethe University Frankfurt, 60438 Frankfurt am Main, Germany. [3]Groningen Institute for Biomolecular Sciences and Biotechnology, University of Groningen, 9747AG Groningen, the Netherlands. [4]Fritz Haber Center for Molecular Dynamics Research, Institute of Chemistry, The Hebrew University of Jerusalem, Jerusalem 9190401, Israel. [5]Department of Biochemistry, University of Cambridge, 80 Tennis Court Road, Cambridge CB2 1GA, UK. [6]European Molecular Biology Laboratory, EMBL Hamburg c/o DESY, 22607 Hamburg, Germany. [7]Institute of Biological Information Processing (IBI-7: Structural Biochemistry), Forschungszentrum Jülich, Jülich, Germany. [8]JuStruct: Jülich Center for Structural Biology, Forschungszentrum Jülich, Jülich, Germany. [9]Univ. Grenoble Alpes, CEA, CNRS, Institut de Biologie Structurale (IBS), 38000 Grenoble, France. [10]European Molecular Biology Laboratory, European Bioinformatics Institute (EMBL-EBI), Wellcome Genome Campus, Hinxton, UK. [11]University Medical Center Göttingen, Institute for Auditory Neuroscience and InnerEarLab, Robert-Koch-Str. 40, 37075 Göttingen, Germany. [12]Cluster of Excellence "Multiscale Bioimaging: from Molecular Machines to Networks of Excitable Cells" (MBExC), University of Göttingen, Göttingen, Germany. [13]Present address: School of Chemistry, University of Edinburgh, Edinburgh EH9 3FJ, UK. [14]These authors contributed equally: E. Podoliak, G. H. U. Lamm. ✉e-mail: alexey.alekseev@med.uni-goettingen.de; kirill.kovalev@embl-hamburg.de

covalently attached to the lysine residue of the helix G (K255 in KR2) via a Schiff base (RSB)[1,9,10]. NDQ rhodopsins form pentamers in the membrane as was shown for KR2[9,11,12]. The transported substrate, sodium, is not bound inside these rhodopsins in the non-illuminated (resting) state, but was found at the oligomerization interface of KR2 coordinated by two neighboring protomers[9,12]. Upon light illumination, NDQ rhodopsins undergo a photocycle with several intermediate states, transitions between which result in sodium translocation across the membrane. Namely, there are the K, L, M, and O intermediates of a typical NDQ rhodopsin, where the O state was shown to be the only one associated with the transient sodium binding inside the protein[1]. For several NDQ rhodopsins, such as KR2 and the NaR from *Indibacter alkaliphilus* (*Ia*NaR), the O state was demonstrated to consist of several sub-states[13–18].

There is still no complete understanding of the molecular mechanism of light-driven sodium pumping by NDQ rhodopsins despite the thorough characterization of KR2. For the latter it was shown that the conformational change in the RSB region takes place upon sodium binding in the O state[14,19]. N112 of the NDQ motif, which points outside of the protomer towards the pentamerization interface in the ground state, flips inside the protein to coordinate the transiently-bound sodium ion[12,19]. This considerable movement (~5 Å in amplitude) is accompanied by the flip of the L74 side chain, creating sufficient space for N112 inside the KR2 protomer[19]. Such a synchronous switch of the N112-L74 pair is believed to be one of the key determinants of light-driven sodium pumping[19]. Surprisingly, while the role of N112 of that pair in the KR2 functioning was studied in detail[20,21], the L74 residue often remained out of the focus despite its involvement in sodium-pumping-associated conformational changes. Nevertheless, it was evidenced by the L74A mutation, which dramatically decreases the sodium-pumping activity of KR2, suggesting its important role in the NDQ rhodopsin[19].

Here, we bioinformatically analyzed the clade of NDQ rhodopsins and found that although the majority of the proteins possess leucine/isoleucine at the position of L74 in KR2, there is a subgroup of proteins bearing glutamate at this position. To see what effect it has on protein structure and function we investigated a rhodopsin from *Erythrobacter sp. HL-111*, *Er*NaR, belonging to this subgroup. *Er*NaR, in contrast to KR2, demonstrated unique spectroscopic properties with only a minor dependence of absorption spectra on pH. Furthermore, *Er*NaR is efficient in pumping sodium with high selectivity over protons even at acidic pH values as low as 5.0. Single particle cryo-electron microscopy (cryo-EM) and X-ray crystallography showed an unusual conformation of the rhodopsin in the resting state with a very short H-bond between E64 and D105 (corresponding to L74 and D116 in KR2, respectively). These findings provide essential information on the mechanisms of NDQ rhodopsins and natural ways of tuning of their functional properties.

## Results and discussion

### Two subgroups of NaRs

Initially, we performed a bioinformatic analysis of the rhodopsins possessing the NDQ motif in available gene databases (UniProtKB, UniParc, GenBank, and MGnify) to obtain the complete list of members of the clade. In total, we identified 351 unique complete sequences of NDQ rhodopsins which resulted in 219 sequences with less than 90% pairwise sequence identity. The proteins cluster into several branches in the phylogenetic tree (Fig. 1a). Closer analysis of these branches showed that, while there are some interesting features of their representatives, the overall clade of NDQ rhodopsins can be divided into two major subgroups, Subgroup 1 and Subgroup 2 (Fig. 1a). Members of the Subgroup 1 possess leucine or isoleucine at the position of L74 in KR2 (Fig. 1b, c). KR2 belongs to Subgroup 1, which is also the larger one. Members of Subgroup 2 possess glutamic acid at the position of L74 in KR2 (Fig. 1b). Importantly, this position is located close to the RSB and

L74 is involved in conformational changes in KR2 associated with sodium translocation[12,19] (Fig. 1c, d). Thus, the introduction of glutamic acid might significantly affect the functional, spectroscopic, and structural properties of the NDQ rhodopsins belonging to Subgroup 2. In addition, although the internal polar and rechargeable residues are conserved within the entire NDQ rhodopsins clade (Fig. 1b, left), the amino acid residues at the surface, including those of the oligomerization interface, are significantly different between the two subgroups (Fig. 1b, right). For instance, the D102 residue, forming the inter-protomeric sodium binding site on the extracellular surface of the KR2 pentamer[9], is either absent or substituted with other amino acid residues in the Subgroup 2 (Fig. 1b, e).

Most of the investigated NDQ rhodopsins belong to Subgroup 1. Our analysis shows that only one protein, a sodium pump from *Salinarimonas rosea* DSM21201 (*Sr*NaR), from Subgroup 2 was studied[6]. *Sr*NaR has been shown to differ from other NaRs. To date, the available data on *Sr*NaR and the members of Subgroup 2 are very limited. Therefore, to study the properties of the members of Subgroup 2, we selected another representative, an NDQ rhodopsin from *Erythrobacter sp.* HL-111 (*Er*NaR), and performed its thorough characterization. *Er*NaR was selected as a promising target for obtaining high-resolution structural data using X-ray crystallography as it lacks unstructured termini and interhelical loops. This allowed us to compare the properties and organization of *Er*NaR with known sodium pumps like KR2.

### Functional characterization of *Er*NaR

To study *Er*NaR in mammalian cells, we cloned a human-codon optimized gene of *Er*NaR to the previously published expression cassette that contained the enhanced yellow fluorescent protein (EYFP), membrane trafficking signal (TS) and endoplasmic reticulum export signal (ES) from potassium channel Kir2.1 and N-terminal part of channelrhodopsin (C2C1)[22]. We transiently expressed *Er*NaR in this construct (hereinafter enhanced *Er*NaR, or e*Er*NaR) in HEK293T cells and evaluated its localization by confocal imaging. Subcellular expression of e*Er*NaR was predominantly confined to the plasma membrane, with low to none at all amounts of protein in intracellular compartments (Fig. 2a).

Next, we used the whole cell patch-clamp technique to study the functional properties of e*Er*NaR. Expecting e*Er*NaR to be an outward sodium or proton pump, we measured the photocurrents at the same extracellular pH (pH$_e$) 7.5 and 110 mM [Na$^+$]$_e$ while varying intracellular pH (pH$_i$) and [Na$^+$]$_i$. Indeed, e*Er*NaR appeared to be an outwardly directed sodium pump, with photocurrent highly dependent on intracellular [Na$^+$]$_i$ (Fig. 2b–d). Notably, e*Er*NaR showed a profound nonlinear voltage dependence at 130 mM [Na$^+$]$_i$, with near-zero currents in negative voltages (Fig. 2c, left). A voltage dependence was also reported in KR2[22], but it was not as pronounced as observed in e*Er*NaR.

To assess the capability of e*Er*NaR to pump H$^+$, for each [Na$^+$]$_i$ we tested three pH$_i$ values (5.0, 7.5, and 9.0) (Fig. 2c). In the presence of sodium in the intracellular solution (20 mM and 130 mM [Na$^+$]$_i$) we observed photocurrents at all studied pH$_i$. A change in pH$_i$ from acidic (5.0) to alkaline (9.0) slightly increased the amplitude of photocurrent (Fig. 2c, left and middle plots). While at 20 mM [Na$^+$]$_i$ this effect was not significant, at 130 mM [Na$^+$]$_i$ the difference reached statistical significance ($P = 0.0349$) when the photocurrents were compared at +60 mV (Fig. 2d). In addition, at 130 mM [Na$^+$]$_i$ we determined the characteristic time of the photocurrent decay after the light was switched off (t$_{off}$, monoexponential fit) in all tested pH$_i$ values (Fig. 2e). Acidic intracellular conditions led to slightly decelerated off-kinetics of e*Er*NaR (statistically not significant).

In the absence of intracellular sodium, e*Er*NaR exhibited immeasurably low photocurrents in all tested pH$_i$. Under similar experimental conditions, when sodium was removed from the intracellular solution, KR2 was shown to pump protons[22]. In e*Er*NaR, the proton currents

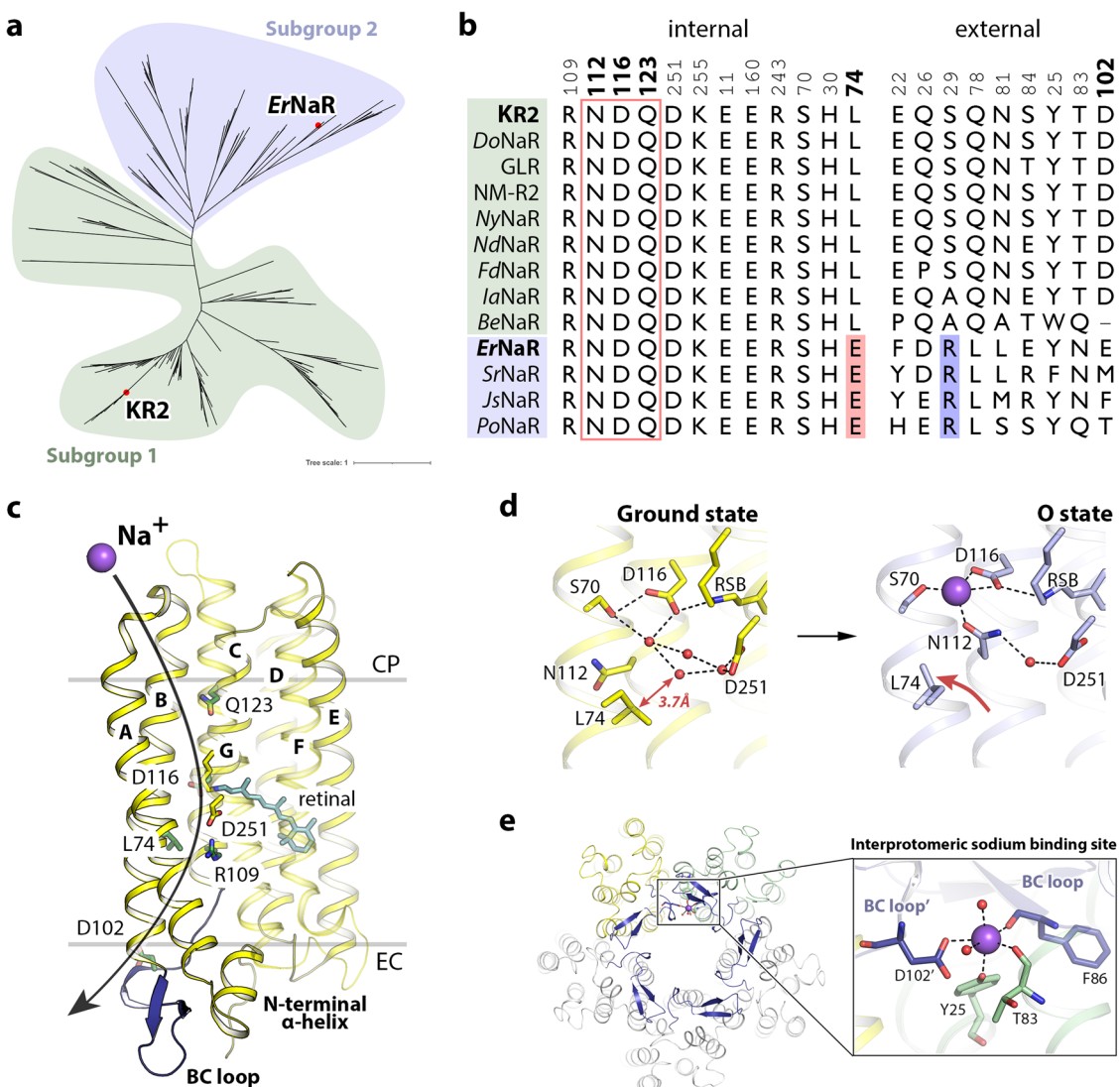

**Fig. 1 | Phylogeny of NDQ rhodopsins. a** Phylogenetic tree of the NDQ rhodopsins clade. Subgroups 1 and 2 are highlighted in green and blue, respectively. **b** Sequence alignment of the key regions of the representative NDQ rhodopsins: internal polar/rechargeable amino acid residues (left), functionally and structurally relevant external amino acid residues (right). For the representative NDQ rhodopsins we selected biophysically characterized proteins also described in[22]. The key positions are marked with bold numbers according to the KR2 sequence. Additional RSB counterion (E64 in *Er*NaR) in the Subgroup 2 is highlighted red. Additional positively charged residue near the interprotomeric sodium binding site (R19 in *Er*NaR) is highlighted blue. **c** Overall side view of the KR2 protomer (PDB ID: 6YC3[19]). Putative sodium translocation pathway is shown with a black arrow. Helices are indicated with bold capital letters (A-G). Hydrophobic/hydrophilic membrane core boundaries are shown with gray horizontal lines. **d** The RSB region of KR2 in the ground (left, yellow, PDB ID: 6YC3[19]) and O (right, blue, PDB ID: 6XYT[19]) states of the photocycle and the role of L74. Distance from L74 to the nearest water molecule in the Schiff base cavity is indicated with a red arrow (left) and is given in bold italic. The flipping motion of the L74 side chain upon sodium binding in the O state is also indicated with a red arrow (right). The sodium ion is shown with the purple sphere. **e** Interprotomeric sodium binding site in KR2 (PDB ID: 6YC3[19]). Overall view of the KR2 pentamer from the extracellular side (left) and detailed view of the site (right). Two neighboring protomers are colored yellow and green. The BC loop is colored dark blue. The sodium ion (purple sphere) coordination is indicated with black dashed lines.

---

were negligible even at pH_i 5.0, suggesting that e*Er*NaR and likely other members of Subgroup 2 of the NDQ rhodopsins are more selective to sodium than representatives of Subgroup 1. Besides, the difference in photocurrent amplitude at +60 mV was statistically significant between 130 mM and 20 mM [Na⁺]_i ($P = 0.0142$, $P = 0.0028$ and $P < 0.0001$ at pH_i 5.0, 7.5 and 9.0, respectively) and between 130 mM and 0 mM [Na⁺]_i ($P = 0.0012$, $P < 0.0001$ and $P < 0.0001$ at pH_i 5.0, 7.5 and 9.0, respectively) (Fig. 2d). However, the difference between 20 mM and 0 mM [Na⁺]_i was not significant in all pH_i, suggesting that e*Er*NaR might require higher intracellular sodium concentrations to successfully function. To verify that *Er*NaR can function in mammalian cells without fused fluorescent protein EYFP, we conducted an additional experiment. We transfected HEK293T cells with C2C1-*Er*NaR-TS-

FLAG mRNA and monitored the localization of the expressed protein by staining the cells with antibody against FLAG-tag. Subcellular expression of C2C1-*Er*NaR-TS-FLAG was mostly restricted to intracellular compartments (Fig. S1a), presumably, due to the lack of endoplasmic reticulum export signal (ES). To label the cells that express *Er*NaR for electrophysiological studies, we cotransfected the cells with two separate mRNAs - C2C1-*Er*NaR-TS-FLAG and EYFP. We kept extracellular conditions identical to our previous experiments (pH_e 7.5 and 110 mM [Na⁺]_e) and measured the photocurrents at pH_i 7.5 and two intracellular sodium concentrations (130 and 0 mM [Na⁺]_i). In the presence of intracellular sodium (130 mM [Na⁺]_i) the photocurrent showed nonlinear voltage dependence, similar to what we report for e*Er*NaR (Fig. S1b, c). However, the amplitude of photocurrent

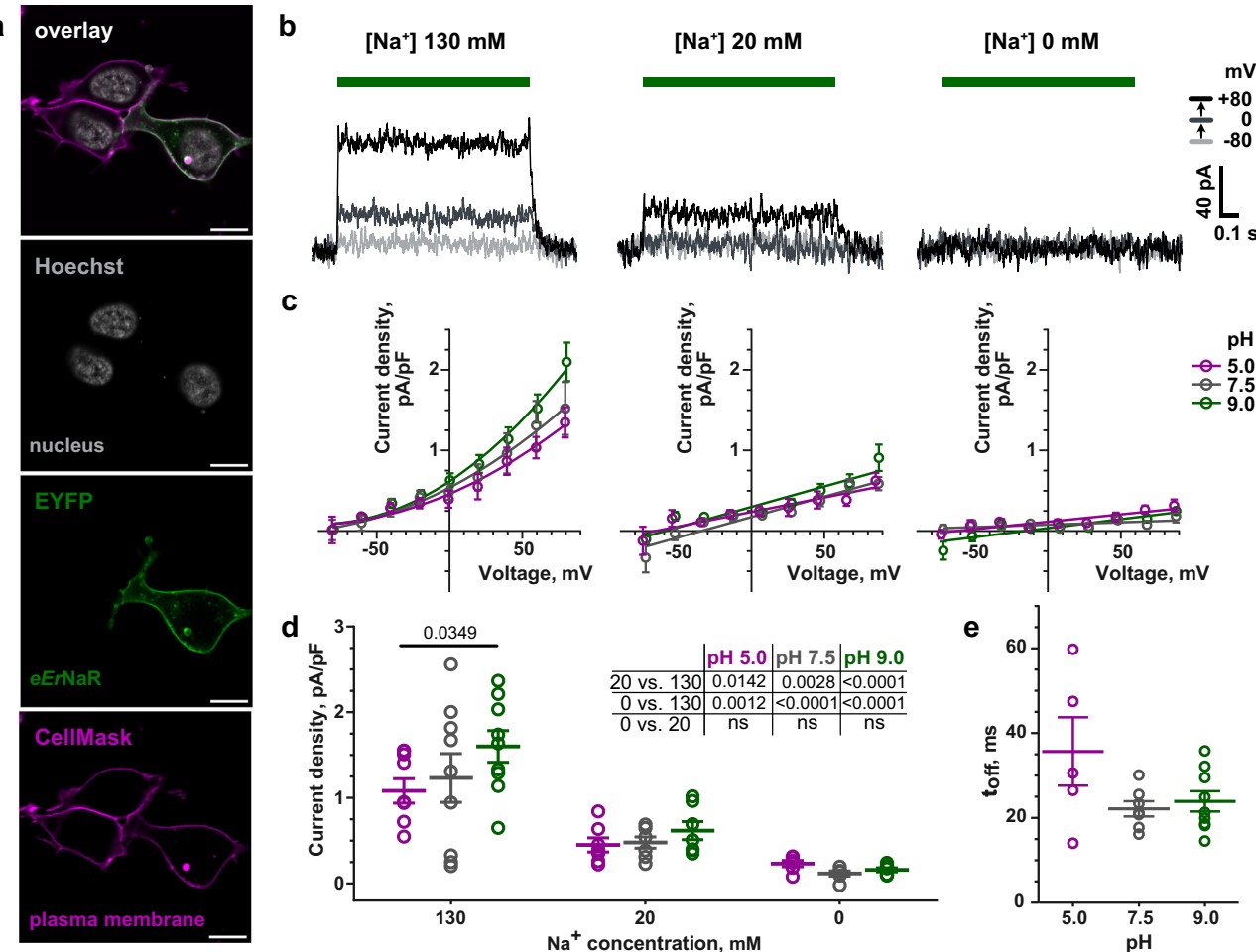

**Fig. 2 | Functional characterization of eErNaR in HEK293T cells. a** Representative confocal images of HEK293T cell expressing eErNaR. Fluorescence of EYFP, fused to eErNaR, is shown in green; the plasma membrane stain CellMask - in magenta; nucleus stain Hoechst 33342 - in gray. The co-localization of EYFP and CellMask appears in white. Scale bars, 10 μm. The localization was similar in at least 6 representative confocal images from 2 independent transfections.
**b** Representative photocurrents of eErNaR recorded from HEK293T cells at 130 mM (left), 20 mM (middle) and 0 mM (right) intracellular $[Na^+]_i$ and $pH_i$ 7.5. **c** Voltage dependence of the stationary photocurrents of eErNaR at intracellular $[Na^+]_i$ 130 mM (left), 20 mM (middle) and 0 mM (right) and different $pH_i$ values (LJP-corrected; normalized to respective cell capacitance; mean ± SEM; n = 9 cells at $[Na^+]_i$ 130 mM, $pH_i$ 5.0, 7.5 and 9.0; n = 7 at $[Na^+]_i$ 20 mM, $pH_i$ 5.0, 7.5, 9.0 and at $[Na^+]_i$ 0 mM, $pH_i$ 9.0; n = 6 at $[Na^+]_i$ 0 mM, $pH_i$ 5.0 and 7.5). **d** Stationary

photocurrents of eErNaR at +60 mV, normalized to respective cell capacitance (mean ± SEM and individual data points). Data were extracted from the recordings at different intracellular $[Na^+]_i$ and $pH_i$ described in (**c**). Normalized currents were analyzed using two-way ANOVA with two Turkey's multiple comparisons tests – for the effect of $pH_i$ at fixed $[Na^+]_i$ (depicted on the graph, ns is not shown) and for the effect of $[Na^+]_i$ at fixed $pH_i$ (inset table). **e** Kinetics of photocurrent decay upon light-off at 130 mM intracellular $[Na^+]_i$ and different $pH_i$ (mean ± SEM and individual data points of n = 5, 7, and 9 cells at $pH_i$ 5.0, 7.5 and 9.0, respectively). The time constant ($t_{off}$) was determined by monoexponential fit of photocurrent decay at holding voltage +80 mV. Data were analyzed using Kruskal-Wallis test with Dunn's multiple comparisons test (all ns). **b**–**e** All patch-clamp experiments were conducted at 110 mM extracellular $[Na^+]_e$, $pH_e$ 7.5; LED light with maximum at 550 nm was applied for 1 s at 34.3 mW/mm² irradiance. ns – not significant.

decreased compared to eErNaR, likely due to the poor membrane targeting of C2C1-ErNaR-TS-FLAG. In the absence of intracellular sodium (0 mM $[Na^+]_i$) the photocurrents of C2C1-ErNaR-TS-FLAG diminished to near-zero values. The observed statistically significant (P = 0.0006) difference in photocurrent amplitudes between 130 mM and 0 mM $[Na^+]_i$ at +60 mV (Fig. S1d) confirms our findings from experiments with eErNaR.

## Spectroscopy of ErNaR

Next, we studied the spectroscopic properties of detergent-solubilized ErNaR. In contrast to KR2[1] and many other microbial rhodopsins[1,23], ErNaR undergoes only a small spectral red-shift of 3 nm upon acidification from pH 8.0 ($λ_{max}$ 535.5 nm) to pH 4.3 ($λ_{max}$ 538.5 nm) (Fig. 3a). An additional 6.5 nm red-shift is observed upon further acidification to pH 2.3 ($λ_{max}$ 545.0 nm) (Fig. S2). The pH titration of the dark state of the ErNaR absorption spectrum yielded two pKa values ($pK_{a,1}$ = 3.34 ± 0.04 (Fig. S2b) and $pK_{a,2}$ = 5.71 ± 0.19

(Fig. S2c)). It should be noted that the detergent-solubilized ErNaR remains pentameric at both pH 2.3 and 8.0 as clearly observed in size-exclusion chromatography (Fig. S3a). Cryo-EM experiments at pH 4.3 and 8.0 also showed the rhodopsin in the pentameric form. Thus, the observed minor spectral shift is the feature of the pentameric form of ErNaR and is not connected to the change of the oligomeric state.

A similar weak pH dependence was found for the ultrafast dynamics of ErNaR. For all measured conditions (pH 8.0 and 4.3), the ultrafast pattern looks similar to what was already shown for other microbial rhodopsins at neutral to alkaline pH values (Fig. 3b; Fig. S4)[23–27]. Namely, after photo-excitation, the excited state decays on a sub-ps timescale and the formation of a hot ground state intermediate at ~610 nm is observed. This intermediate is commonly termed as J and relaxes to form the first stable red-shifted photoproduct $K_1$ at ~590 nm described by the lifetime distribution in the 1-4 ps range (Fig. S4a).

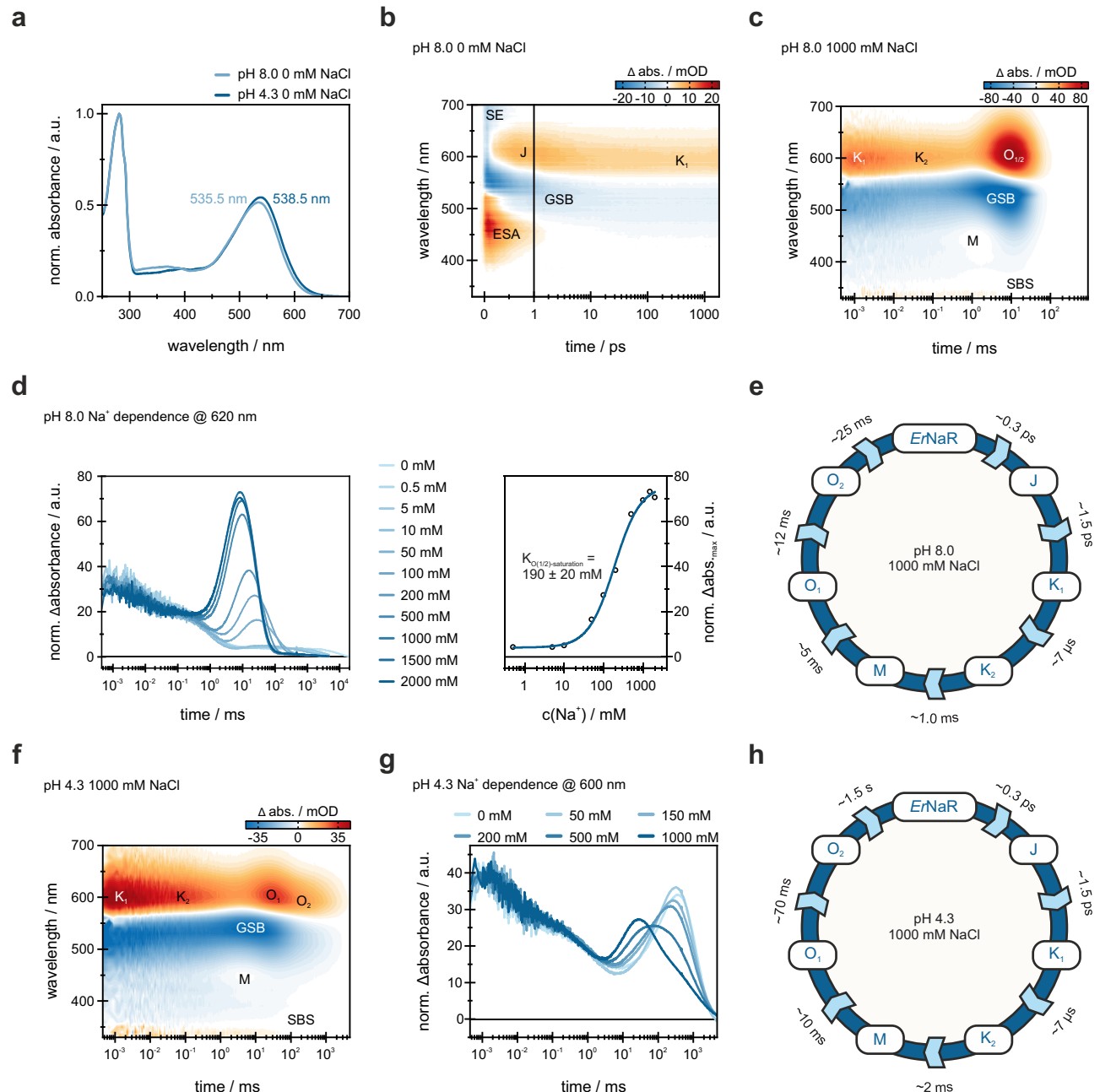

**Fig. 3 | Spectroscopic characterization of *Er*NaR. a** Normalized absorption spectra of *Er*NaR at acidic (pH 4.3) and neutral (pH 8.0) conditions. The spectra have been normalized to the absorbance at 280 nm. **b** 2D-contour plot of a fs-TA measurement of *Er*NaR (pH 8.0, 0 mM NaCl). The timescale is linear until 1 ps and logarithmic afterwards. The signal amplitude is color coded as follows: positive (red), no (white) and negative (blue) abs. The mentioned abbreviations represent excited-state absorption (ESA), ground-state bleach (GSB) and stimulated emission (SE). **c** 2D-contour plot of a flash photolysis measurement of *Er*NaR (pH 8.0, 1000 mM NaCl). Due to spectral similarity, the region of the $O_1$ and $O_2$ intermediates is indicated as $O_{1/2}$. The SBS abbreviation represents the "second bright state" signal at 335 nm being a marker of the 13-*cis* retinal configuration[30,31]. **d** Sodium dependence of the transient at 620 nm, indicative of the $O_1$ and $O_2$

intermediates, at pH 8.0. Transients have been normalized to have the same Δabs. at 0.1 ms. The absorption maximum of the $O_1$ and $O_2$ intermediates was plotted against the sodium concentration and the resulting data was fitted with the Hill equation to obtain the sodium dependence of the $O_1$ and $O_2$ intermediates. **e** Schematic model of the *Er*NaR photocycle at pH 8.0, 1000 mM NaCl. **f** 2D-contour plot of a flash photolysis measurement of *Er*NaR (pH 4.3, 1000 mM NaCl). The SBS abbreviation represents the "second bright state" signal at 335 nm being a marker of the 13-*cis* retinal configuration. **g** Sodium dependence of the transient at 600 nm, representative for the $O_1$ and $O_2$ intermediates, at pH 4.3. Transients have been normalized to have the same Δabs. at 0.1 ms. **h** Schematic model of the *Er*NaR photocycle at pH 4.3, 1000 mM NaCl.

We also studied the photocycle kinetics of *Er*NaR on the ns to s timescale under various conditions using flash photolysis. At pH 8.0, the photocycle, as well as the impact of different sodium concentrations on the *Er*NaR, are similar to what was observed for KR2[1,28,29]. The $K_1$-intermediate at the end of the ultrafast measurement (1.8 ns), remains present at the beginning of the flash photolysis time scale

(450 ns), showing that $K_1$ is populated throughout the ns timescale and no photocycle intermediate is missed due to the experimental time gap (Fig. 3b, c). $K_1$ decays within the early μs-range (lifetime distribution in the range of 5–10 μs) to form the $K_2$-intermediate. This intermediate state is considered to be K-like because of the more pronounced red-shift compared to the known L-intermediates of other

microbial rhodopsins[28,29]. The transition to the blue-shifted M-intermediate then occurs within ~1 ms. In the absence of sodium, M is the last observed intermediate of the photocycle and the protein subsequently relaxes back to the parent state, represented in the lifetime distributions centered at 5-7 s (Fig. S5a). In the presence of a sufficient amount of sodium ions, however, the duration of the M state is significantly shortened and, as was observed also for KR2[1], a strong signal assigned to the red-shifted O-intermediate rises (Fig. 3d (left)) within 2–10 ms (Fig. S5b, c). Comparison of the maximum signal intensity of the O-states for different NaCl concentrations allowed the determination of $K_{O(1/2)-saturation} = 190 ± 20$ mM for ErNaR, demonstrating that high sodium concentrations are required to observe the "sodium-pumping mode" (Fig. 3d). Thus, at 100 mM NaCl, ErNaR shows mixed kinetics of one subpopulation likely undergoing the "sodium-pumping mode" photocycle while the other subpopulation undergoes a long-living M-intermediate before directly relaxing back to the parent state (Fig. S5b). In the case of the sodium-pumping regime, analysis of the flash-photolysis data showed the presence of two O intermediates ($O_1$ and $O_2$) similar to those shown for KR2 and IaNaR[14,15] (Fig. 3e, Fig. S5c; see "Methods" for more details on the data analysis). The $O_1$-to-$O_2$ transition coincides with the decay of the signal at 335 nm (Fig. 3c; Fig. S5c). This near-UV band was previously termed as second bright state (SBS) and was shown to be a marker of the 13-cis retinal configuration in KR2[30] and inward proton pump NsXeR[31]. Thus, our data suggest that the $O_1$ state of ErNaR contains 13-cis retinal, while $O_2$ has an all-trans retinal. These results are in line with the findings and interpretation of Fujisawa et al.[17] for IaNaR.

At pH 4.3, the photocycle kinetics of ErNaR differs from that at pH 8.0 (Fig. 3f; Fig. S6). The kinetics of the early photocycle intermediates up to the formation of the M state under all tested conditions are similar to that observed at pH 8.0 at 1000 mM NaCl (Fig. S5c, S6). The M state shows a weak absorbance and a fast decay (Fig. S6), which corresponds to the reprotonation of the RSB and can be explained by the higher proton concentration at acidic conditions and was also observed for KR2[32].

However, in contrast to KR2, the $O_1$ and $O_2$ states were observed for ErNaR at pH 4.3 and are clearly sensitive to sodium concentrations (Fig. S6). Indeed, our flash photolysis data on KR2 at pH 4.3 showed no rise of the O state even at high sodium concentrations, such as 1000 mM NaCl (Fig. S7). In ErNaR at pH 4.3, the two O-states are present even under sodium-free conditions, likely reflecting proton uptake and binding close to the RSB at the late stages of the photocycle due to the higher proton concentration. In general, at pH 4.3, the spectra of both O states are slightly blue-shifted compared to those at pH 8.0 (Fig. S5, S6, S8). With increasing sodium concentration, the kinetics of the two O-intermediates are influenced differently, leading to a spectral and temporal separation of $O_1$ and $O_2$ at 1000 mM NaCl at pH 4.3 (Fig. 3f,g). Namely, the $O_2$ state is blue-shifted compared to the $O_1$ intermediate (Fig. 3f). To exclude possible ionic strength effects, we performed measurements with 1000 mM KCl and 1000 mM N-Methyl-D-glucamine (NMG), an organic monovalent cation that is commonly used to replace sodium ions in electrophysiological experiments[33]. The measurements at pH 4.3 with 1000 mM KCl showed a photocycle very similar to the one obtained for 0 mM and 100 mM NaCl at pH 4.3 (compare Fig. S6a, b and Fig. S9b) lacking the clear spectral differences observed for 1000 mM NaCl (Fig. S6c and S9b). With 1000 mM NMG at pH 4.3, the photocycle is very similar to the one with 1000 mM KCl and 0 mM NaCl at pH 4.3. The obtained lifetime densities for 1000 mM KCl and 1000 mM NMG are in good agreement as well (compare Fig. S6a and S9). Therefore, this allowed us to rationalize that under acidic conditions the observed separation of the $O_1$ and $O_2$ states in response to the increase of sodium concentration is a direct result of the sodium binding close to the RSB. Similar to pH 8.0, the SBS signal indicates that a 13-cis configuration of the retinal is suggested for $O_1$ and the all-trans configuration for $O_2$ (Fig. 3f).

Taken together, while at pH 8.0 the behavior of ErNaR is similar to KR2 and other studied NaRs, at pH 4.3 it demonstrates unique spectral features. First, the spectral shift upon the pH decrease in ErNaR is small, being <10 nm between pH 8.0 and 2.3 and only 3 nm between pH 8.0 and 4.3. We suggest that these small spectral shifts of ErNaR are not connected to the protonation of the main RSB counterion, D105, since its protonation is expected to cause much larger spectral changes[1,34,35]. Indeed, in KR2, the protonation of D116 upon pH decrease from 8.0 to 4.3 results in the red-shift of ~25 nm[1]. In addition, the ultrafast kinetics of ErNaR are pH-independent (Fig. S4), arguing for the absence of the D105 protonation at acidic pH. Thus, we speculate that the minor red-shift of the ErNaR spectrum at low pH is associated with the protonation of the rechargeable residues distant from the RSB. It could be also connected to the partial redistribution of the charges at the RSB counterion complex including residues D105, E64, and D242 as described further in the manuscript.

The second feature of ErNaR at low pH is the presence of the O-states in the photocycle as well as their sensitivity to sodium. We suggest that the preservation of the O-states supports our above-mentioned hypothesis on the deprotonated form of the RSB counterion D105 even at pH 4.3. The sensitivity of the O-states to sodium ions is also in line with the observed sodium-transport activity of ErNaR at low pH (Fig. 2C, D). We speculate that the blue-shift of the O-states at pH 4.3 compared to pH 8.0 has several reasons. Specifically, at low sodium concentrations (0 mM and 100 mM) the O-states likely reflect the binding of a proton in the active center of ErNaR due to much higher concentration of protons than that of sodium. This hypothesis is in line with the model of competitive uptake of protons and sodium ions shown for KR2[32]. However, as the proton-pumping activity of ErNaR was not detected at low pH we suggest that after the binding the proton is released back to the cytoplasm in the same manner it is proposed for another sodium pump GLR[3]. Thus, at low sodium concentrations, the spectral and kinetic differences between the O-states of the ErNaR photocycles at pH 4.3 and 8.0 might originate from the binding of different substrates. We speculate that at high sodium concentration (1000 mM) the sodium ion is bound in the active center of ErNaR in the O-states in the same manner at pH 4.3 and 8.0. In this case, the slight blue-shift of the O-states at pH 4.3 might be caused by the protonation of the residues distant from the RSB upon acidification.

## Cryo-EM structure of the pentameric ErNaR

In order to investigate the molecular basis of the unique functional and spectral properties of ErNaR and the effect of Leu to Glu replacement in rhodopsins, we used single-particle cryo-electron microscopy (cryo-EM) to determine the structure of the protein in different conditions. We obtained the structures of ErNaR at acidic (4.3) and neutral (8.0) pH values at the resolution of 2.50 and 2.63 Å, respectively (Fig. S10). Cryo-EM maps allowed us to model residues 1-272 of ErNaR. The overall organization of the pentamer is similar to that of KR2 and other MRs (Fig. 4a; Fig. S11). Namely, the nearby protomers interact via helices A and B and BC loop containing a β-sheet (Fig. S11).

At the same time, the concavity in the central part of the pentamer at the extracellular side is organized differently in ErNaR and KR2 (Fig. 4b). In general, while in KR2 this region is polar and interacts with numerous water molecules identified using X-ray crystallography (PDB ID: 6YC3[19]), in ErNaR it is more hydrophobic (Fig. 4b). Moreover, the cryo-EM maps reveal the presence of fragments of hydrophobic chains in this region of ErNaR (Fig. 4b).

Notably, the cryo-EM data strongly suggest that the inter-protomeric sodium binding site found in the KR2 pentamer[9] is absent in ErNaR (Fig. 4c). Although the maps show the spherical density at a position similar to that of the sodium ion in KR2 (Figs. 1d, 4c), the coordination is unlikely to favor sodium binding. In KR2 the sodium binding site is formed by two neighboring protomers; namely, by the

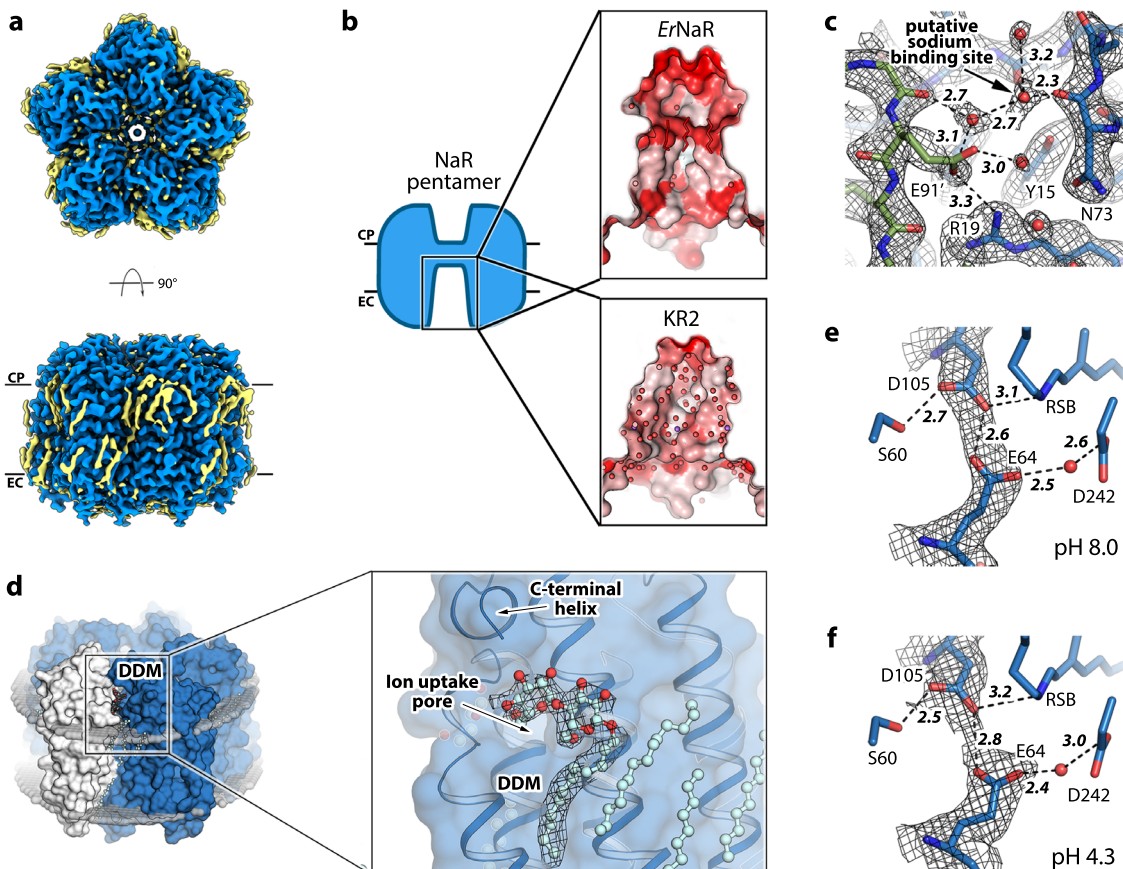

**Fig. 4 | Cryo-EM structures of *Er*NaR. a** Overall view of the *Er*NaR pentamer from the extracellular (top) and along the membrane plane (bottom). **b** Comparison of the hydrophobicity of the concave aqueous basin in the center of the *Er*NaR and KR2 pentamers. **c** The region of interprotomeric sodium binding site (according to the KR2 structure) absent in *Er*NaR. The cryo-EM map (pH 8.0) is shown with black mesh. **d** The DDM molecule in the cryo-EM structures of *Er*NaR. 5 molecules of DDM are located between neighboring protomers at the cytoplasmic side of the protein. The polar head of DDM is located next to the entrance to the ion uptake cavity and interacts with the C-terminal helix found in *Er*NaR. **e** The central region of the *Er*NaR protomer at pH 8.0. The cryo-EM map is shown with black mesh and indicates direct interaction between D105 and E64. **f** The central region of the *Er*NaR protomer at pH 4.3. The cryo-EM map is shown with black mesh and indicates direct interaction between D105 and E64. The lengths of the H-bonds are given in Å and shown with bold italic numbers.

side chains of D102′ and Y25 as well as carbonyl oxygens of T83 and F86 and two water molecules (Fig. 1d). In *Er*NaR, E91 (corresponding to D102 in KR2) is located 2.5 Å further from the position of the probable sodium binding site and thus is unlikely to coordinate the ion (Fig. 4c). Moreover, E91 directly interacts with the positively charged R19, located within only 5.5 Å from the possible sodium binding site (Fig. 4c). The end of the BC loop of *Er*NaR is also arranged slightly differently from that of KR2 (Fig. S11). The sequence alignment also indicates that the region of the interprotomeric sodium binding site is organized differently in Subgroups 1 and 2 of NDQ rhodopsins (Fig. 1b, right). Thus, we speculate that the interprotomeric sodium binding site found in KR2 is likely a feature of only Subgroup 1 but not Subgroup 2 of NDQ rhodopsins.

Another feature of *Er*NaR is the presence of a detergent (n-Dodecyl-beta-Maltoside, DDM) molecule in the cleft between the rhodopsin protomers (Fig. 4d). The DDM molecule is found at the cytoplasmic leaflet of the membrane and its polar head is located near the pore in the *Er*NaR surface likely serving as the entrance for sodium ions (Fig. 4d). The polar head also interacts with the C-terminus of the protein (Fig. 4d). Nevertheless, the DDM molecule does not block the pore.

Although the quality of the cryo-EM map is sufficient to place the side chains of amino acid residues as well as protein-associated water molecules, it is still limited and does not allow us to precisely determine the distances between the functional groups of rhodopsin. This hampers the understanding of the molecular mechanisms of *Er*NaR.

For instance, while the cryo-EM data clearly show a direct interaction between the D105 residue of the characteristic NDQ motif with the E64 residue of *Er*NaR at both pH 4.3 and 8.0, the map regions corresponding to these residues merge together and individual positions cannot be resolved (Fig. 4e, f). Nevertheless, this might be indicative that there is a rather short H-bond between these two residues.

### The internal organization of the *Er*NaR protomer
In order to resolve details of the internal organization of *Er*NaR, we crystallized the protein using *in meso* approach and determined its structures at pH 4.6 and 8.8 at 1.7 Å resolution using X-ray crystallography. The crystals originally appeared at pH 4.6 and contained a monomer of *Er*NaR in the asymmetric unit. To obtain high-resolution structure of *Er*NaR at high pH, we soaked the crystals in the buffer solution with pH 8.8 (see Methods for details). The structures of the rhodopsin at pH 4.6 and 8.8 appeared nearly identical (RMSD of 0.06 Å). The structure of the *Er*NaR protomer obtained using X-ray crystallography is also very similar to that in the pentameric cryo-EM structure (RMSD of overall structures of 0.5 Å). Since the crystal structure is similar to that obtained with cryo-EM but reveals more details on the internal organization of *Er*NaR, we used it for the further analysis.

The overall protomer organization of *Er*NaR is similar to that of KR2 (Fig. 5a; Fig. S11c, d) with a large cavity at the cytoplasmic part of both *Er*NaR and KR2, likely acting as an ion uptake cavity (Fig. 5a,b). In the central region, D105 (D116 in KR2) is directly H-bonded to the RSB

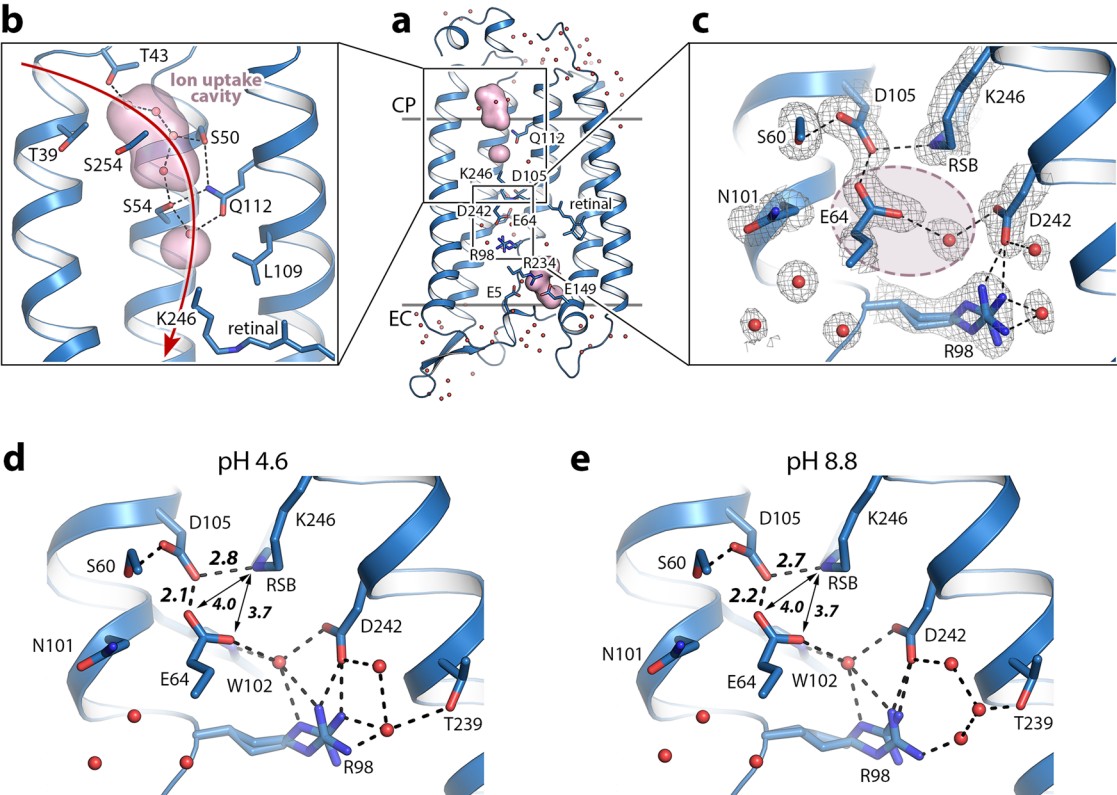

**Fig. 5 | Crystal structures of *Er*NaR. a** Overall side view of the *Er*NaR monomer. Hydrophobic/hydrophilic membrane core boundaries are shown with gray lines. The cytoplasmic (CP) and extracellular (EC) sides of the membrane are indicated. **b** Detailed view of the cytoplasmic internal part of *Er*NaR. Cavities were calculated using HOLLOW and are shown with a pink surface. Red arrow indicates putative sodium uptake pathway. **c** The RSB region of *Er*NaR at pH 4.6. 2Fo-Fc electron density maps are contoured at the level of 1.0σ and are shown with black mesh.

H-bonds are shown with black dashed lines. Light-red area indicates the region of the Schiff base cavity found in KR2 and absent in *Er*NaR. **d** The detailed view of the RSB region of *Er*NaR at pH 4.6. **e** The detailed view of the RSB region of *Er*NaR at pH 8.8. H-bonds are shown with black dashed lines. The lengths of the key H-bonds between the RSB and D105 and D105 and E64 are given in Å and shown with bold italic numbers. The distances between the oxygen atoms of E64 and the RSB are shown with black arrows and are given in Å.

(Fig. 5c). The retinal cofactor is in the all-*trans* configuration in the resting state of *Er*NaR (Fig. 5a). The internal extracellular region of *Er*NaR is separated from the RSB region with R98 (analog of R109 in KR2) and comprises numerous polar residues including the E5-E149-R234 triad, organized almost identically to the E11-E160-R243 triad of KR2 (Fig. 5a).

Despite some similarities, the protomer structure of *Er*NaR differs from that of KR2 in several aspects. First, the N-terminus is shorter in *Er*NaR and the N-terminal α-helix has only one turn compared to two turns in KR2 (Fig. S11c). Nevertheless, the N-terminus is capping the inside of *Er*NaR (Fig. 5a, S11c). Also, an additional small C-terminal α-helix was found in *Er*NaR, comprising residues 267-272 of the protein (Fig. S11d).

Second, the cytoplasmic ends of helices D, E, and F are shifted by more than 3 Å in *Er*NaR compared to those of KR2 (Fig. S11d). This might be related to the notable differences in the length and organization of the CD and EF loops of the two rhodopsins (Fig. S11d). However, these shifts of the helices are not reflected in the internal organization of the cytoplasmic part of *Er*NaR. In particular, the ion uptake cavities of *Er*NaR and KR2 are highly similar.

Some differences between KR2 and *Er*NaR were also found in the extracellular side of the proteins. For example, there is a leucine (L68) in *Er*NaR at the position of Q78 in KR2, which was shown to be critical for sodium release from the latter rhodopsin[19]. However, the Q78L mutation of KR2 has no effect on sodium pumping; therefore, we suggest that this natural substitution found in *Er*NaR is in line with the earlier suggested mechanism of sodium release from the NDQ rhodopsins[19].

## Role of the E64 residue in the active center of *Er*NaR

Another noteworthy difference between *Er*NaR and KR2 was found in the RSB region. E64, a unique characteristic residue of *Er*NaR and Subgroup 2 of NDQ rhodopsins, is pointed towards the RSB and interacts directly with D105 (D116 in KR2) (Fig. 5c). As a result, a large Schiff base cavity found in the ground state of KR2 in the pentameric form is absent in *Er*NaR and only one water molecule is found in this region (Fig. 5a, c). It is coordinated by E64, R98, W102, and D242 (Fig. 5c, d). At the same time, the N101 side chain (N112 in KR2) is pointed towards the oligomerization interface similar to that found in the ground state of pentameric KR2[12] (Fig. 5c, d).

The interaction between D105 and E64 has not been observed before and is an intriguing feature of *Er*NaR. Significantly higher resolution of the X-ray crystallography data (1.7 Å vs 2.5 Å for the cryo-EM data) allowed us to accurately position the side chains of E64 and D105 at both pH 4.6 and 8.8 (Fig. 5c). The distance between the nearest oxygens of E64 and D105 was only 2.1 and 2.2 Å at pH 4.6 and 8.8, respectively, which was extremely short for a donor-acceptor distance in a normal H-bond and strongly suggests a low-barrier H-bond (LBHB) between the residues (Fig. 5d, e). The E64 side chain was also connected to the second aspartic acid counterion of the RSB, D242, through a water-mediated H-bond chain (Fig. 5c–e).

In order to get insights into the protonation states of carboxylic residues in close vicinity to the RSB and to gain more information on the strong interaction between E64 and D105 we performed hybrid quantum mechanics/molecular mechanics (QM/MM) simulations based on the crystal structure of *Er*NaR. In particular, the protonation state of the E64 residue is of high interest as this residue is only found

in Subgroup 2 of NDQ rhodopsins. Thus, we probed various protonation states of the counterion complex composed of E64, D105, and D242. For each of the three carboxylic acids we have considered one of four proton orientations as shown in Fig. S12. In total, 61 protonation patterns were considered (1 with zero-protonation, 12 single-protonation, and 48 double-protonation). For each protonation pattern an energy minimization was performed, followed by the calculation of vertical excitation energies (Fig. S12a).

We found that the spectral and structural properties of _Er_NaR can be explained by system 6 with two protonated carboxylic acids, namely, E64 and D242, because the protonation patterns with zero or one protonated carboxylic acid have too high excitation energies (Fig. S12a, b). The algebraic diagrammatic construction (ADC(2)) method that was used for excitation energy calculations has an estimated error of 0.1–0.3 eV from benchmark studies[36]. Only the structures with a double-protonated counterion complex fall within this range. Within this subset of protonations, model 6 has the lowest ground state energy. Besides the match between the computed and measured excitation energies, also the agreement with the crystal structure supports the model 6, since the protonation of D105 would break the D105-RSB salt bridge, which is clearly observed in the high-resolution structures. Thus, we conclude that in agreement with the spectroscopic and structural data on _Er_NaR, D105 remains deprotonated in a wide range of pH values, while E64 is likely protonated even at a pH as high as 8.8. In the QM/MM-refined structure (Fig. S12d) the distance between D105 and E64 is 2.5 Å. In order to analyze the nature of the interaction we performed the energy decomposition analysis using Zero-Order Symmetry Adapted Perturbation Theory (SAPT0). The SAPT0 analysis shows a significant repulsion, but it was overcompensated by electrostatics, polarization, and dispersion interactions (Fig. S12b).

Thus, our structural, spectroscopy, and QM/MM data strongly suggest that the E64 residue in the helix B of _Er_NaR likely maintains low $pK_a$ of the main RSB counterion D105. To probe this hypothesis, we performed a mutational analysis of _Er_NaR. We substituted E64 with Leu and Gln and studied spectroscopic properties of the E64L and E64Q variants of the rhodopsin. The KR2-like E64L mutant showed a notable red shift of 50 nm upon pH decrease from 11 to 6, supporting our hypothesis on the key role of E64 for the spectral stability of _Er_NaR (Fig. S13a). Surprisingly, the shift is ~2 times larger than that of KR2 (~25 nm) and also occurs at higher pH value. This indicates that E64 is not the only determinant of the spectral differences between _Er_NaR and KR2. We speculate that the above-described differences in other regions of the two proteins also contribute to their spectral differences.

The E64Q mutant should mimic the _Er_NaR with protonated E64 and thus is expected to show similar spectral behavior to that of the wild type (WT) protein. However, it demonstrated a larger spectral shift upon pH titration (~20 nm vs. ~10 nm in E64Q and WT, respectively) (Fig. S13b). The maximum absorption wavelength at neutral and alkaline pH values is also slightly different in the mutant (~525 nm) and WT (~535 nm) (Fig. S13b). We suggest that the spectral differences between the mutant and the WT protein originate from the absence of the LBHB between D105 and Q64 in _Er_NaR-E64Q since it cannot be formed between the Asp and Gln residues. This result additionally supports our hypothesis on the LBHB between D105-E64 and its role in the $pK_a$ lowering of D105 in _Er_NaR.

In summary, we propose the following mechanism of the E64 influence on the pKa of D105 with at least two components: (1) an unusually short H-bond to the protonated E64 affecting the electrochemical properties of D105; (2) the tight integration of the E64 side chain into the H-bond network of the RSB region additionally stabilizes the overall conformation associated with the deprotonated form of D105. The LBHB between E64 and D105 means that the E64-D105 pair might also share a proton in the ground state of _Er_NaR in a similar way

that shown for the proton release group of BR[37]. In this case, only a minor spectral red-shift of _Er_NaR upon pH decrease (Fig. S2) might be connected to the charge redistribution with the counterion complex including the D105-E64 pair reflected in different lengths of proposed LBHB at pH 4.6 and 8.8 (2.1 and 2.2 Å, respectively) (Fig. 5d, e). Lastly, the RSB-E64 distance of only 3.7 Å (Fig. 5d, e), together with the direct effects of the E64Q and E64L mutations on the absorption spectrum (Fig. S13) and its pH sensitivity of the protein, validate our assignment of E64 as an additional RSB counterion in _Er_NaR. Since the E64 is conserved within the Subgroup 2 of NDQ rhodopsins we also suggest the same role of the glutamic acid in other members of this subgroup.

## Comparison of the two subgroups of NaRs

The bioinformatic analysis of the clade of NDQ rhodopsins revealed two major subgroups. Our functional, spectroscopical, and structural data on _Er_NaR from Subgroup 2 presented here and the previously reported data on KR2 as well as other NDQ rhodopsins of Subgroup 1 allowed us to compare the properties and molecular mechanisms of these two subgroups of NaRs.

Electrophysiology demonstrated that _Er_NaR is capable of active transport of sodium across the membrane in a wide range of pH values, including acidic pH as low as 5, which has not been shown for KR2 or other members of Subgroup 1 of NDQ rhodopsins. Thus, the functional data available on KR2 in literature does not allow the direct comparison of its properties to _Er_NaR. Nevertheless, in KR2, as well as in other members of Subgroup 1, the pH decrease leads to the protonation of the main RSB counterion, aspartic acid of the characteristic NDQ motif[1,3]. This protonation correlates with the deceleration of the photocycle and loss of the O-states reflecting the lowered sodium binding efficiency[1,32]. On the contrary, _Er_NaR lacks such pH dependence on absorption spectra and retains the sodium-dependent O-states even at pH 4.3. Therefore, the cumulative data allow us to suggest that the presence of an additional carboxylic residue in close proximity to the RSB in the rhodopsins of Subgroup 2 might contribute to their efficient functioning as sodium pumps in a wider range of pH values than that of the Subgroup 1 members. Further comparative studies of the NDQ rhodopsins from both subgroups are required to validate this hypothesis.

Our kinetics studies of _Er_NaR demonstrated that the competitive uptake of sodium and protons might be a common feature of Subgroup 1 and Subgroup 2 of NDQ rhodopsins. Clearly, the acceleration of the M-state decay upon the increase of either sodium or proton concentration shows that both types of ions can be uptaken by _Er_NaR in a competitive manner, similar to what was found for KR2[32]. The uptake of protons at low pH is also nicely represented by the presence of the O-states in the _Er_NaR photocycle under sodium-free conditions at pH 4.3.

While the conformation of the RSB region in the resting state is different in _Er_NaR and KR2, there is a similarity in the orientation of N101 and N112 of the characteristic NDQ motif in _Er_NaR and KR2, respectively (Fig. 6). The asparagine is oriented outside of the protein protomer towards the pentameric interface in the resting state of both rhodopsins. For KR2, this conformation was directly demonstrated in 2019 but was originally proposed back in 2016 and was named "expanded" as it is characterized by a large water-filled cavity near the RSB[12,38]. The expanded conformation was only found at neutral pH values, when the rhodopsin works as a sodium pump[12]. Thus, this conformation was assigned to the functional form of the protein. In _Er_NaR, the conformation of the RSB region lacks the large Schiff base cavity but with respect to the N101 orientation is similar to the expanded one of KR2. This allows us to conclude that in the resting state of all NDQ rhodopsins under sodium-pumping conditions, including both Subgroups 1 and 2, the common characteristic feature is the orientation of asparagine of the functional motif outside of the protomer towards the oligomerization interface. We named this

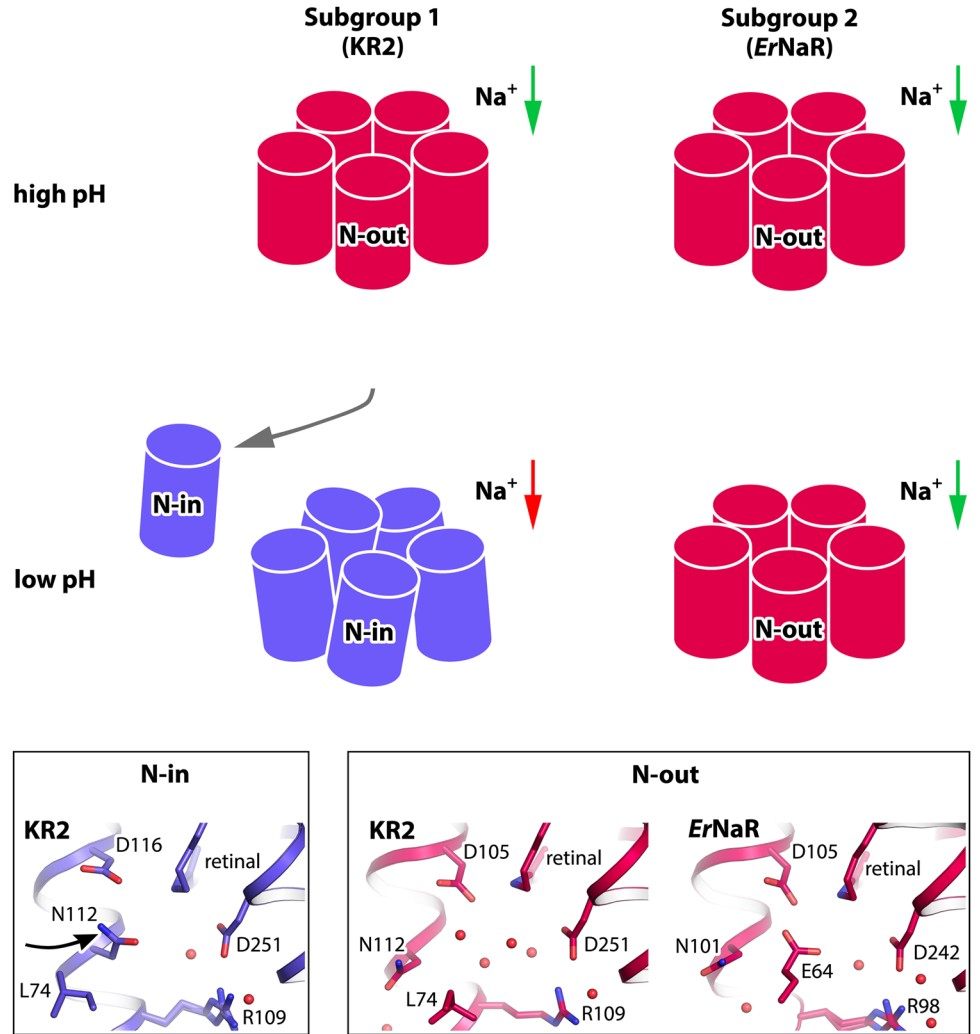

**Fig. 6 | N-in and N-out conformations of the NDQ rhodopsins.** The scheme of the difference in pH dependence of the spectral properties (spectral shift upon acidification in Subgroup 1 is shown with the change of the color from pink to blue), the pentamer stability (the disturbed pentameric assembly at low pH is shown for KR2 schematically by disoriented protomers; partial dissociation of the pentamers into protomers is also indicated by an additional gray arrow), and the conformation between Subgroup 1 and 2 of NDQ rhodopsins (N-in and N-out conformations of the active centers). Green arrows indicate sodium transport across the membrane by NaRs. Red arrow indicates the absence of sodium transport. The detailed view of the RSB regions of KR2 and ErNaR in N-in and N-out conformations is given in the bottom panels. Black arrow indicates different orientation of Asn residue of the NDQ motif in N-in and N-out conformations.

common conformation as the "N-out" one (Fig. 6). Consequently, the conformation of NDQ rhodopsin with the asparagine oriented inside the protomer should be named "N-in" (Fig. 6).

As shown for KR2, the N-out conformation is provided by the pentameric assembly[12]. The N112 side chain is H-bonded to the residues of the nearby rhodopsin molecule when flipped outside of the protomer. In pentamers of KR2 at low pH, only the N-in conformation was found[12]. Furthermore, in the monomer, only the N-in conformation was observed at all pH values[12] (Fig. 6). In contrast to KR2, ErNaR adapts the N-out conformation in the pH range of 4.3-8.8 in both monomeric and pentameric forms (Fig. 6). Thus, we suggest that the conformations of the residues comprising the characteristic NDQ motif (N101, D105, Q112 in ErNaR and N112, D116, Q123 in KR2) are similar in the ground states of both proteins, which is likely a common feature of the members of both Subgroups 1 and 2. At the same time, this N-out conformation seems to be more stable in Subgroup 2 than in Subgroup 1, and is independent of pH and oligomeric state.

Although the pentameric assembly is a common feature of all NDQ rhodopsins, the central areas of the pentamers of the members of Subgroups 1 and 2 are organized differently. The interprotomeric sodium binding site found in KR2 and thought to be present in all proteins belonging to Subgroup 1, is likely absent in ErNaR and the rest of Subgroup 2. The environment of the concave aqueous basin in the middle part of the ErNaR oligomer is more hydrophobic than that of KR2. The sodium release through this central region and a possible relay mechanism involving the surface-bound interprotomeric sodium ion in KR2 were proposed to lower the energetic barriers for the ion translocation against strong electrochemical gradients[19]. Thus, we suggest that sodium-release pathways and mechanisms might be different in Subgroups 1 and 2 of the NDQ rhodopsins.

Constantly expanding gene databases allowed us to discover the naturally occurring variations within the NDQ rhodopsin clade of light-driven sodium pumps. Our findings demonstrated a natural way of fine-tuning the active sodium transporters by the introduction of an additional carboxylic residue in close proximity to the retinal Schiff base. It results in (1) efficient sodium pumping at a wide range of pH values, (2) spectral stability, and (3) high ion selectivity, which could make the members of the newly highlighted subgroup of NDQ rhodopsins more useful for biotechnological applications. While the mechanistic insights on the resting state reported here shed more light on the NaRs organization, further structural investigations of their intermediate states are required to understand their molecular

mechanisms. According to the reported results, there might be multiple mechanisms of light-driven sodium pumping by NDQ rhodopsins. Our study may be the basis for the upcoming time-resolved crystallography and/or cryo-EM as well as cryotrapping studies of the NDQ rhodopsins.

## Methods

### Search for and phylogenetic analysis of the NDQ rhodopsins

To retrieve the NDQ rhodopsin genes from available sequence databases, jackhmmer search (N iterations = 5) was performed using the sequence of KR2 rhodopsin (UniProt ID: N0DKS8) as a template against the UniProtKB (2023-09-15), UniParc (2023-09-15), Genbank (2023-09-15), and MGnify (2023-09-15) databases. The 6-letter motifs for each protein of the output multiple sequence alignment (MSA) file, corresponding to the amino acid residues at the positions of R109, N112, D116, Q123, D251, and K216 of KR2 were retrieved using a custom Python script using jupyter notebook. The sequences possessing the RNDQDK motif were selected, filtered to contain more than 250 amino acid residues, and re-aligned separately using MUSCLE[39]. Duplicated sequences were removed using a custom Python script. The selected sequences were also manually inspected to contain full seven transmembrane α-helices. The maximum likelihood phylogenetic tree was built using IQ-TREE webserver[40] (Number of bootstrap alignments: 1000) and visualized using iTOL (v6.8)[41]. For the building of the phylogenetic tree all sequences with identity higher than 90% were removed using CD-HIT software[42]. The full MSA file, 90% identity-filtered MSA file, and phylogenetic tree file are provided as Supplementary files. Two subgroups of NDQ rhodopsins were separated by using the presence/absence of the glutamate of the position of L74 in KR2 as a criterion.

### Cloning and mRNA production

*Er*NaR (Uniprot ID: A0A1H1XA63) coding DNA was optimized for *E.coli* or human codons using GeneArt (Thermo Fisher Scientific). Genes were synthesized commercially (Eurofins). For protein expression and purification, the pET15b plasmid with 6xHis-tag at the C-terminal was used. The E64L and E64Q mutations were introduced using site-directed mutagenesis with primer pairs CACTGCTGCTGGGT-CAGCTGTGG / CTGCGCTAACCATAACAACTGCTGAAATATTTGC and CACTGCAACTGGGTCAGCTGTGG / CTGCGCTAACCATAACAACTGC TGAAATATTTGC, respectively. For electrophysiological recordings of e*Er*NaR, a human codon optimized gene of *Er*NaR was cloned into the pcDNA3.1(-) vector between *Bam*HI and *Hind*III sites together with an N-terminal part of channelrhodopsin (C2C1), membrane trafficking signal (TS) and ER export signal (ES) from potassium channel Kir2.1 and enhanced yellow fluorescent protein (EYFP). C2C1 and TS-EYFP-ES were amplified from pEYFP-N1-eKR2[22], which was a gift from Peter Hegemann (Addgene plasmid #115337). The final construct (C2C1-*Er*NaR-TS-EYFP-ES) was verified by sequencing. The plasmid is available from Addgene (#209851).

For electrophysiological recordings of C2C1-*Er*NaR-TS-FLAG, mRNAs of C2C1-*Er*NaR-TS-FLAG and EYFP were generated. First, C2C1-*Er*NaR-TS-FLAG and EYFP DNA fragments were amplified, using primers with T7 promoter at the 5' end and FLAG-tag sequence at the 3' end (only in case of C2C1-*Er*NaR-TS-FLAG) (Table S1), Phusion™ High-Fidelity DNA Polymerase (Thermo Fisher Scientific) and pcDNA3.1(-)-C2C1-*Er*NaR-TS-EYFP-ES plasmid as a matrix. PCR products were checked on agarose gel electrophoresis, purified with QIAquick gel purification kit (Qiagen) and used as a template for in vitro transcription with HiScribe® T7 ARCA mRNA Kit (with tailing) (New England Biolabs). ARCA capped mRNAs were treated with DNase I and poly(A) tailed with Poly(A) Polymerase (both from the described above kit). mRNAs were purified using the Monarch RNA cleanup kit (New England Biolabs) and stored at −80 °C.

### HEK293T cell culture and transfection

The HEK293T cells (Thermo Fisher Scientific) were cultured at 37 °C with 5% $CO_2$ in high-glucose Dulbecco's Modified Eagle Medium (DMEM), containing 1 mM sodium pyruvate and 1X GlutaMAX, supplemented with 10% heat-inactivated fetal bovine serum, 100 U/ml penicillin and 100 µg/ml streptomycin (all Thermo Fisher Scientific). The cells were regularly tested and were free from mycoplasma. For electrophysiological recordings and confocal imaging cells were seeded on poly-l-ornithine-coated 24-well plates at a concentration of $1.5 \times 10^5$ cells per well. For electrophysiological recordings and live cell confocal imaging of e*Er*NaR, after 24 h, the cells were transiently transfected using 1.5 µg of Polyethylenimine MAX (PEI) (Polysciences) and 1 µg of plasmid per well in Opti-MEM (Thermo Fisher Scientific), supplemented with 1 µM of all-*trans*-retinal (Sigma). The transfection medium was replaced after 4–5 h with full culture medium, supplemented with 1 µM of all-*trans*-retinal. For electrophysiological recordings of C2C1-*Er*NaR-TS-FLAG, 24 h after seeding the cells were cotransfected with mRNAs of C2C1-*Er*NaR-TS-FLAG (0.75 µg) and EYFP (0.25 µg), using 0.75 µl of Lipofectamine™ MessengerMAX™ Reagent (Thermo Fisher Scientific) per well, according to manufacturer's instructions. For confocal imaging of C2C1-*Er*NaR-TS-FLAG, the transfection was performed similarly, but using only 0.75 µg of C2C1-*Er*NaR-TS-FLAG mRNA. Lipofectamine and mRNA complexes were mixed in Opti-MEM (Thermo Fisher Scientific) and added directly to full culture medium, supplemented with 1 µM of all-*trans*-retinal (Sigma). 16 h post transfection the cells were plated on poly-l-ornithine-coated 12-mm glass coverslips or 8-well µ-Slide (ibidi) at 20% confluence and analyzed by whole cell patch-clamp or confocal imaging 6-8 h later.

### Staining and confocal imaging

For live cell imaging HEK293T cells were stained with 5 µg/ml Hoechst 33342 to visualize nuclei and CellMask Deep Red (1:2000, Thermo Fisher Scientific) to label plasma membranes in Opti-MEM solution (Thermo Fisher Scientific). Imaging was performed at 37˚C and 5% $CO_2$ within 30 min after the staining. For immunostaining of C2C1-*Er*NaR-TS-FLAG cells were fixed in 4% paraformaldehyde in PBS and blocked in PBS, containing 2.5% BSA and 0.5% Triton X-100 for 30 min at RT. To visualize the localization of C2C1-*Er*NaR-TS-FLAG, the cells were incubated with AlexaFluor 488-conjugated anti-FLAG-tag antibody (1:100, MA1-142-A488, Thermo Fisher Scientific) in PBS, supplemented with 2.5% BSA and 0.1% Triton X-100 for 2 h at RT. Following two PBS washes, the cells were stained with 5 µg/ml Hoechst 33342 (Thermo Fisher Scientific) for 5 min at RT to label nuclei. The cells were then washed with PBS again and stored in PBS at +4 ˚C (no longer than 2 weeks).

Imaging was performed on a Leica Stellaris 8 confocal microscope at the Microscopy Core Facility of the Medical Faculty at the University of Bonn. AlexaFluor 488, EYFP and CellMask Deep Red were excited with a tunable White Light Laser (440 – 790 nm) at wavelengths 499, 514, and 653 nm, respectively. Hoechst 33342 was excited with a 405 nm laser. Middle-plane images were acquired using 63×/ 1.2 NA water immersion objective and additional ×3 zoom at 1024 × 1024 pixels and with line averaging of 8. The resulting images were processed in FIJI. The confocal images were obtained from at least 6 representative cells for each construct (from 2 independent transfections).

### Patch-clamp recordings, data processing and statistics

Whole-cell patch clamp recordings were performed at room temperature, using a Multiclamp 700B amplifier (Molecular Devices). The signals were filtered at 10 kHz and digitized at sampling rate of 20 kHz with an Axon Digidata 1550B digitizer (Molecular Devices) using Clampex 11 Software (part of pCLAMP 11, Molecular Devices). Patch pipettes (3-6 MΩ) were fabricated from borosilicate glass with filament (GB150F-8P, Science Products GmbH) on a horizontal puller (Model P-1000, Sutter Instruments). Light was provided by a pE-800 system

(CoolLED), controlled via TTL input, and connected to the optical path of Olympus SliceScope Pro 6000 upright microscope (Scientifica) via pE-Universal Collimator (CoolLED). LED light with maximum at 550 nm was applied at 34.3 mW/mm² irradiance in the focal plane of the 20×/ 0.5 NA water objective to activate $Er$NaR. To calculate the irradiance the output light power was measured using a bolometer (Coherent OP-2 VIS, Santa Clara) and divided by the illuminated area. The intensity distribution at the illuminated spot was fitted with gaussian and a radius, within which 95% of total power accumulates, was used as a measure of the spot size. The reference electrode was connected to the bath solution via an agar bridge with 150 mM KCl. The series resistance was <20 MΩ. Each cell was recorded three times and averaged directly in Clampex 11 Software to improve the signal-to-noise ratio. The ionic composition of the extracellular solution and all the intracellular solutions is indicated in Table S2. Data were corrected for the respective liquid junction potential (LJP) after recording. LJPs for all solutions were measured directly and are stated in Table S2. Data were analyzed using custom Wolfram Mathematica scripts and GraphPad Prism software. Photocurrent amplitudes were normalized to respective cell capacitance. Photocurrents at +60 mV were calculated from the linear fit of data points in positive voltages after LJPs correction. Time constant ($t_{off}$) was determined by monoexponential fit of photocurrent decay upon light-off at holding voltage +80 mV. The data are presented as mean ± SEM of N = 6-9 cells for the photocurrent amplitudes or N = 5-8 cells for off-kinetics. The data from individual cells are also shown when appropriate. Photocurrents of e$Er$NaR at +60 mV were tested for normal distribution using Shapiro-Wilk normality test (passed) and analyzed using two-way ANOVA with two Tukey's multiple comparisons tests – for the effect of $pH_i$ changes at fixed $[Na^+]_i$ and for the effect of $[Na^+]_i$ changes at fixed $pH_i$. $t_{off}$ was analyzed using Kruskal-Wallis test with Dunn's multiple comparisons test. Photocurrents of C2C1-$Er$NaR-TS-FLAG at +60 mV were analyzed using Mann-Whitney test.

### Protein expression, solubilization, and purification

$E.coli$ cells were transformed with pET15b plasmid containing the gene of interest. Transformed cells were grown at 37 °C in shaking baffled flasks in an autoinducing medium ZYP-5052[43], containing 10 mg/L ampicillin. They were induced at an $OD_{600}$ of 0.6–0.7 with 1 mM isopropyl-β-D-thiogalactopyranoside (IPTG). Subsequently, 10 μM all-$trans$-retinal was added. Incubation continued for 3 h. The cells were collected by centrifugation at 5000 × $g$ for 20 min. Collected cells were disrupted in an M-110P Lab Homogenizer (Microfluidics) at 25,000 p.s.i. in a buffer containing 20 mM Tris-HCl, pH 8.0, 5% glycerol, 0.5% Triton X-100 (Sigma-Aldrich), and 50 mg/L DNase I (Sigma-Aldrich). The membrane fraction of the cell lysate was isolated by ultracentrifugation at 125,000 × $g$ for 1 h at 4 °C. The pellet was resuspended in a buffer containing 20 mM Tris-HCl, pH 8.0, 0.2 M NaCl and 1% DDM (Glycon) and stirred overnight for solubilization. The insoluble fraction was removed by ultracentrifugation at 125,000 $g$ for 1 h at 4 °C. The supernatant was loaded on a Ni-NTA column (Qiagen), and the protein was eluted in a buffer containing 20 mM Tris-HCl, pH 8.0, 0.2 M NaCl, 0.4 M imidazole, and 0.1% DDM. The eluate was subjected to size-exclusion chromatography on a Superdex 200i 300/10 (GE Healthcare Life Sciences) in a buffer containing 20 mM Tris-HCl, pH 8.0, 0.2 M NaCl and 0.05% DDM. In the end, protein was concentrated to 70 mg/ml for crystallization and stored at −80 °C.

### Steady-state absorption spectroscopy and pH titration

Absorption spectra of $Er$NaR samples were measured with an absorption spectrometer (Specord600, Analytik Jena). Before and after each experiment, absorption spectra were taken to check sample quality. For the pH titration, samples were prepared to have a protein concentration of ~0.4 mg/mL. The protein was suspended in the titration buffer containing 10 mM potassium citrate, 10 mM MES, 10 mM HEPES, 10 mM Tris, 10 mM CHES, 10 mM CAPS, and 100 mM Arg-HCl. The pH was adjusted with tiny amounts of [5000 mM] HCl or [5000 mM] KOH, respectively.

### Ultrafast transient absorption spectroscopy

Ultrafast transient absorption measurements were performed with a home-built pump-probe setup. A fs laser system - consisting of an Amplifier (Spitfire Ace-100F-1K, Spectra-Physics), seeded by a Ti:Sapphire oscillator (Mai Tai SP-NSI, Spectra-Physics) and pumped by a Nd:YLF laser (Empower 45, Spectra-Physics) - was used as the source for ultrashort laser pulses (100 fs, 800 nm, 1 kHz repetition rate). A home-built two-stage noncollinear optical parametric amplifier (NOPA) was used to generate the pump pulses at 550 nm. The white light continuum pulses used to probe the absorption changes of the sample were generated by focusing the 800 nm laser fundamental into a $CaF_2$-crystal (3 mm). For the detection of the pump-probe signals a spectrometer (AvaSpec-ULS2048CL-EVO-RS, Avantes) was used. The measurements were performed in a 1 mm quartz cuvette and the sample was adjusted to have a protein concentration of ~3.8 mg/mL protein. The sample was continuously moved in a plane perpendicular to the excitation beam to avoid photo-degradation. The excitation pulses were adjusted to an energy of 90 nJ/pulse.

### Transient flash photolysis spectroscopy

A Nd:YAG laser (SpitLight 600, Innolas Laser) was used to pump an optical parametric oscillator (preciScan, GWU-Lasertechnik). The OPO was set to generate excitation pulses with a central wavelength of 550 nm and an average pulse energy of ~2.2 mJ/cm². As probe light sources a Xenon or a Mercury-Xenon lamp (LC-8, Hamamatsu) were used. Two identical monochromators (1200 L/mm, 500 nm blaze), one in front and one after the sample, were used to set the chosen probing wavelengths. Absorption changes were detected by a photomultiplier tube (Photosensor H6780-02, Hamamatsu) and converted into an electrical signal afterwards. This signal was recorded by two oscilloscopes (PicoScope 5244B/D, Pico Technology) with overlapping timescales. For each transient 30 acquisitions were measured and averaged to increase the S/N ratio. To obtain data files with a reasonable size for further analysis, raw data files were reduced using forward averaging and a combined linear and logarithmic timescale.

Detergent-solubilized samples were measured in a 2 × 10 mm quartz cuvette and prepared to have a concentration of ~0.8 mg/mL protein. For conditions pH 8.0 0, 100, 1000 mM NaCl, pH 4.3 0, 100, 1000 mM NaCl, pH 4.3 1000 mM KCl and pH 4.3 1000 mM NMG, absorption changes were measured between 330 and 700 nm with a stepsize of 10 nm. Additional steps of the performed $Na^+$ titration were measured by recording transients at characteristic wavelengths (pH 8.0: 340 nm, 450 nm, 540 nm, 610 nm and 620 nm; pH 4.3: 340 nm, 450 nm, 540 nm, 580 nm, 600 nm and 620 nm). For the KR2-WT measurements, characteristic transients of the respective photocycle intermediates (340 nm, 410 nm, 530 nm, 550 nm, 600 nm and 630 nm) were investigated at sodium concentrations 200 mM, 500 mM and 1000 mM NaCl.

### Analysis of time-resolved spectroscopic data

Analysis of time-resolved spectroscopic data was performed using OPTIMUS software[44], available free-of-charge at [www.optimusfit.org]. The data of the ultrafast transient absorption and transient flash photolysis measurements were objected to the model-free lifetime distribution analysis (LDA) yielding the lifetime distributions of the individual photointermediate transitions, which are summarized in a lifetime distribution map (LDM). The lifetimes of the photocycle intermediate transitions have been determined at the maximum of the respective lifetime distribution in the corresponding LDM and are therefore mentioned as approximate numbers (Fig. S14).

To determine the LDMs visualized in the manuscript, a Tikhonov regularization with 200 regularization factors in the range of 0.01–5 were used together with 200 lifetimes to calculate the inverse Laplace transform of the experimental dataset $S(\tau, \lambda_{exc}, \lambda_i)$ as explained in more detail in[44]. The start lifetime was set to 0.03 ps for the measurements on the ultrafast timescale and 0.0004 ms for measurements on the ns to s timescale. The end lifetime was set to 5400 ps for the ultrafast timescale and three times the last experimental time point for the ns to s timescale. The shown LDMs have been chosen via the L-curve criterion.

### Cryo-EM grid preparation and data collection
For cryo-EM all samples were originally purified in the Buffer 1 (20 mM Tris pH 8.0, 200 mM NaCl, 0.05% DDM) and concentrated to 60 mg/ml using 100,000MWCO concentrator. For the structure of *Er*NaR at pH 8.0 (pH 8.0 dataset), the sample was diluted later with Buffer 1 (pH 8.0 structure). For the structure of *Er*NaR at pH 4.3 (pH 4.3 dataset), the sample was diluted with Buffer 2 (100 mM sodium acetate pH 4.3, 200 mM NaCl, 0.05% DDM) and run through Superdex200i 300/10 column in Buffer 2 to remove excess of Buffer 1. After that, the sample was again concentrated to 60 mg/ml 100,000MWCO concentrator. For grid preparation, all samples were diluted to 7 mg/ml, and volume of applied onto freshly glow-discharged (30 s at 5 mA) Quantifoil grids (Au R1.2/1.3, 300 mesh) at 20 °C and 100% humidity and plunged-frozen in liquid ethane. The cryo-EM data were collected using 300 keV Krios microscope (Thermo Fisher), equipped with Gatan K3 detector.

### Cryo-EM data processing
All steps of data processing were performed using cryoSPARC v.4.0.2[45] (Fig. S10). Motion correction and contrast transfer function (CTF) estimation were performed with default settings for all three datasets. Initial volume for template picking was generated after picking particles from pH 8.0 dataset using Topaz[46] pre-trained model, followed by two rounds of 2D classification and ab initio model generation with 1 class, and homogeneous refinement. After that, for all datasets the final set of particles was picked using this volume as a template, with a template picker (150 Å particle radius), followed by duplicate removal with 50 Å distance.

For the pH 4.3 dataset, picked particles were extracted with 3x binning (384 px to 128 px). An initial set of particles was cleaned using two rounds of 2D classification (first round: 80 classes, 80 iterations, 5 final iterations, batch size 400, use clamp-solvent: true; second round: 40 classes, 40 iterations, batch size 200, use clamp-solvent: true). After that, particles were cleaned using a "3D classification" (ab initio model generation with 5 classes, followed by heterogeneous refinement), producing 555,294 particles for the non-binned refinement. These particles were re-extracted with 320 px box size, followed by homogeneous refinement (C5 symmetry, with per-particle CTF and defocus refinement) and local refinement (C5 symmetry, map generated with "Volume tools" at threshold 0.48), yielding a final resolution of 2.50 Å.

For the pH 8.0 dataset, picked particles were extracted with 4x binning (512 px to 128 px). An initial set of particles was cleaned using two rounds of 2D classification (first round: 50 classes, 40 iterations, batch size 200, use clamp-solvent: true; second round: 20 classes, 40 iterations, batch size 200, use clamp-solvent: true). After that, particles were cleaned using a "3D classification" (ab initio model generation with 5 classes, followed by heterogeneous refinement), producing 437,017 particles for the non-binned refinement. These particles were re-extracted with 640 px box size, followed by homogeneous refinement (C5 symmetry, with per-particle CTF and defocus refinement) and local refinement (C5 symmetry), yielding a final resolution of 2.63 Å. The overall view of the cryo-EM maps is shown in Supplementary Movie 1 and 2.

### Model building and refinement
Automatically sharpened maps from cryoSPARC were aligned using UCSF ChimeraX[47]. The pentameric model of *Er*NaR was generated using Alphafold[48] and docked as a rigid body into cryo-EM maps manually in ChimeraX. Further refinement was performed using Phenix[49,50] and Coot[51], producing the final statistics described in Table S3. Visualization and structure interpretation were carried out in UCSF Chimera[47,52] and PyMol (Schrödinger, LLC).

### Crystallization
The crystals of *Er*NaR were grown with an *in meso* approach[53], similar to that used in our previous works[12,19]. In particular, the protein at 70 mg/ml solubilized in DDM in the crystallization buffer was mixed with premelted at 42 °C monoolein (MO, Nu-Chek Prep) in a 3:2 ratio (lipid:protein) to form a lipidic mesophase. The mesophase was homogenized in coupled syringes (Hamilton) by transferring the mesophase from one syringe to another until a homogeneous and gel-like material was formed.

Then, 150 nl drops of a protein–mesophase mixture were spotted on a 96-well LCP glass sandwich plate (Marienfeld) and overlaid with 400 nL of precipitant solution by means of the NT8 or Mosquito crystallization robots (Formulatrix and SPT Labtech, respectively). The best crystals were obtained with a protein concentration of 20 mg/ml (in the water part of the mesophase). The best crystals were obtained using 0.1 M Sodium acetate pH 4.6, 10% PEG550MME (Hampton Research) as a precipitant. The crystals were grown at 22 °C and appeared in 1 month.

For the determination of the *Er*NaR crystals structure at pH 4.6, once the crystals reached their final size, crystallization wells were opened, and drops containing the protein-mesophase mixture were covered with 100 μl of the precipitant solution. For the data collection, harvested crystals were incubated for 5 min in the precipitant solution. For the determination of the *Er*NaR crystals structure at pH 8.8, the crystals were originally grown at pH 4.6 using the same precipitant as described above. Once the crystals reached their final size, crystallization wells were opened and the crystals were soaked for 48 hours with exchanging the buffer three times. Crystals were harvested using micromounts (Mitegen, USA), flash-cooled, and stored in liquid nitrogen.

### Diffraction data collection and treatment
X-ray diffraction data of both structures of the ground state of *Er*NaR at pH 4.6 and 8.8 were collected at the P14 beamline of PETRAIII (Hamburg, Germany) using an EIGER X 16 M and EIGER2 X 16 M CdTe detectors. The data collection was performed using MxCube2 software. Diffraction images were processed using XDS[54]. The reflection intensities were scaled and merged using the Staraniso server[55]. There is no possibility of twinning for the crystals. In both cases, diffraction data from a single crystal were used. The data collection and treatment statistics are presented in Table S4.

### Crystal structure determination and refinement of *Er*NaR
Initial phases for the ground state of monomeric *Er*NaR at pH 4.6 were successfully obtained in the P6122 space group by molecular replacement using MOLREP[56] from the CCP4 program suite[57] using the 4XTL structure of monomeric KR2[9] as a search model. The initial model was iteratively refined using REFMAC5[58] and Coot[59]. The phases for the structure of *Er*NaR at pH 8.8 were determined using MOLREP with the phases of the *Er*NaR structure at pH 4.6 as a search model. The structure refinement statistics are presented in Table S4.

### Molecular simulation
The computational model of *Er*NaR is based on X-ray crystallographic structure with PDB ID 8QLE, reported in this work. The protonation states were determined using propKa online service[60]. The protons

were added using pdb4amber program from AmberTools package[61]. To remove bad contacts energy minimization using the ff14SB AMBER force field[62] was carried out.

The protonation states of the residues E64, D105, and D242 are determined from all possible combinations of their protonation states. We refer to them as protonation patterns. There is only 1 pattern where E64, D105, and D242 are deprotonated (zero-protonation). However, in case that one carboxylic acid side chain is protonated there are four different initial orientations to place the proton (Fig. S12c). Therefore, 12 structures were considered for one protonation of these three side chains (single-protonation) and 48 structures for two protonated side chains (double-protonation). Each structure of a specific protonation pattern was energy minimized using the hybrid quantum mechanics/molecular mechanics (QM/MM) method[63,64].

The QM/MM geometry optimization was performed using the L-BFGS[65] algorithm. The protonated retinal Schiff base including part of the lysine sidechain is part of the QM partition. The QM-MM boundary is placed between the Cε-Cδ position. In addition, the side chains of the residues E64, D105, D242, and a water molecule were part of the QM region. In this case, the QM-MM boundary is put at the Cα-Cβ bond. The MM partition comprises the remaining protein. During the geometry optimization all side chains within 5 Å distance of the retinal were flexible to move, while the rest was frozen. The QM part was treated using the B3LYP[66] density functional and Resolution-of-Identity[67] approximation, with Grimme[68] dispersion correction for the geometry optimization. The correlation-consistent[69] cc-pVDZ atomic basis set and the corresponding auxiliary basis set were used. The protein was treated using the ff14SB AMBER force field[70]. The QM and MM partitions interact through electrostatic embedding. All energy minimizations were performed with the Orca v5.0.3[71] computational package.

The vertical excitation energies were computed for each optimized protonation pattern using the RI second-order Algebraic Diagrammatic Construction scheme (RI-ADC(2)) for the QM part. The RI-ADC(2) calculations of vertical excitation energies were performed with Turbomole7.3 computational package[72]. Protein preparation was done using AMBER tools[62].

### Reporting summary

Further information on research design is available in the Nature Portfolio Reporting Summary linked to this article.

## Data availability

The data that support this study are available from the corresponding authors upon request. Atomic models built using X-ray crystallography and cryo-EM data have been deposited in the Protein Data Bank (PDB) under accession codes 8QR0 (pentameric form at pH 4.3), 8QQZ (pentameric form at pH 8.0), 8QLE (monomeric form at pH 4.6), and 8QLF (monomeric form at pH 8.8). The cryo-EM maps have been deposited in the Electron Microscopy Data Bank (EMDB) under accession codes EMD-18610 (pH 4.3) and EMD-18609 (pH 8.0). Publicly available in Protein Data Bank structures of KR2 under accession codes 4XTL, 6YC3, and 6XYT were used for analysis. The QM/MM optimized structures of model number 1-7 (Fig. S12a) are available in the Zenodo repository [https://doi.org/10.5281/zenodo.10824456]. Source data are provided with this paper.

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

## Acknowledgements

This research was supported by the German Research Foundation (CRC 1507 – Membrane-Associated Protein Assemblies, Machineries and Supercomplexes; Project 05 to J.W. and CRC 1078 – Protonation Dynamics in Protein Function; Project C6 to I.S.). E.M., A.S. and A.G. thank NeCEN personnel for the support during data collection. The access to NeCEN facilities was funded by the Netherlands Electron Microscopy Infrastructure (NEMI), project number 184.034.014 of the National Roadmap for Large-Scale Research Infra- structure of the Dutch Research Council (NWO). A.G. was supported by NWO grant OCENW.KLEIN.141. We thank the Microscopy Core Facility of the Medical Faculty at the University of Bonn for providing support and instru- mentation funded by the Bundesministerium für Bildung und Forschung (BMBF, Federal Ministry of Education and Research) – ACCENT: För- derung von Advanced Clinician Scientist im Bereich Immunopathoge- nese und Organdysfunktion, Gehirn und Neurodegeneration – Förderkennzeichen: 01EO2107. V.B. acknowledges support by the Volkswagen Foundation (Freigeist—A110720) and the Deutsche For- schungsgemeinschaft (EXC-2151-390873048-Cluster of Excellence— ImmunoSensation2 at the University of Bonn). I.S. acknowledges fund- ing from the Israel Science Foundation (Research Center, grant no. 3131/20) and the GIF NEXUS No. I-1560-207.9/2023. K.K. has been supported by EMBL Interdisciplinary Postdoctoral Fellowship (EIPOD4) under Marie Sklodowska-Curie Actions Cofund (grant agreement number 847543). The work of A.A. was supported by funding from the German Research Foundation via the Multiscale Bioimaging - Cluster of Excellence (EXC 2067/1-390729940).

## Author contributions

K.K. performed bioinformatics analysis of the NDQ rhodopsins with the help of A.A. and under the supervision of A.B.; K.K. cloned the *Er*NaR gene for *E.coli* expression; C.B. and K.K. expressed and purified *Er*NaR in *E.coli*; T.B. supervised the expression and purification; E.P. cloned the *Er*NaR gene for electrophysiology studies with the help of N.M.; E.P. did the electrophysiology and confocal microscopy of *Er*NaR under the supervision of A.A. and V.B.; G.H.U.L. and A.V.S. did the spectroscopy of *Er*NaR with the help of M.A. under the supervision of J.W.; K.K. crystal- lized the protein; R.A. helped with initial crystallization trials; V.G. supervised initial crystallization trials; K.K. collected and processed X-ray crystallography data and solved the crystal structures of *Er*NaR; G.B. helped with the X-ray crystallography data analysis; A.S. prepared grids and performed initial characterization and analysis of EM data; E.M. and A.G. processed cryo-EM data and obtained initial pentameric models of *Er*NaR; K.K. refined the cryo-EM structures; A.G. supervised grid preparation, cryo-EM data collection and processing and structure refinement and validation; D.A.F. performed QM/MM simulation and analyzed the results under the supervision of I.S.; V.G. suggested the concept of short strong H-bonds for $pK_a$ control of functional residues; K.K., E.P., A.A., G.H.U.L., D.A.F., E.M. analyzed the data and prepared a manuscript with contributions of V.G., A.G., V.B., T.R.S. and all other authors.

## Funding

## Competing interests

The authors declare no competing interests.
