## [Peer Review File · Nature Communications]

A subgroup of light-driven sodium pumps with an additional Schiff base counterionReviewer #1 (Remarks to the Author):

In their manuscript „A novel subgroup of light-driven sodium pumps with an additional Schiff base counter ion“, Podolaki, Lamm et al. perform a comprehensive biophysical analysis of a new sodium-pumping microbial rhodopsin with an interesting Schiff base counter ion. To point this out upfront: the presented analysis is as comprehensive as one may be. The manuscript reports on patch-clamp measurements, time-resolved UV/Vis measurements, both cryoEM and crystal structures at two different pH values each and high-level QM/MM calculations. I myself am a theoretical biophysicist and therefore can only comment on the reliability of the QM/MM calculations, but I definitively support a publication in Nature Communications.

My comments:

-- I would like the authors to prepare a subpanel directly in Fig. 1 with a structure of the protein that sketches the assumed sodium pumping path, the involved crucial residues and names the different helices. All this information is partitioned over the remaining figures and supplementary figures, but it would be very helpful to have this information available directly at the start.

-- Concerning the QM/MM calculation method section: The method description is a bit short to understand what you were actually doing. Were the QM/MM simulations all performed only in Orca? This is not completely clear to me from the description in the methods. What starting structures did you use, cryoEM as well as crystal structures? Did you try different input structures? If yes, did they converge? What was the exact simulation approach, did you perform QM/MM dynamics or only structural minimizations?

-- Fig. S9: If the “GS energy” in subpanel A is a potential energy, then model 6 has the highest, not the lowest energy. Did you maybe multiply the numbers with an erroneous factor of -1 ? Furthermore, the different protonation states require the addition of one or two protons into your box. That means you have additional (beneficial) proton-electron interactions in the Hamiltonian that lower the potential energy of the system. Isn't the observation that the “-1” states have the lowest GS energy compared to “-2” and “-3” then trivial, because you deal here with an open system instead of a closed one? How robust are your estimates of the absorption maximum in nm, and how large is the error bar in comparison to experiment based on the method and level of theory you use?

Reviewer #2 (Remarks to the Author):

Podoliak et al. identified a new subgroup (subgroup 2) of light-driven sodium pumps (NDQ rhodopsins) bearing in addition to the two known aspartic acid residues a glutamic acid in close vicinity to the retinal Schiff base (RSB). The authors selected ErNaR from *Erythrobacter* sp. HL-111 as representative member of subgroup 2, and performed extensive functional and structural studies. Based on the obtained and presented data, the authors suggest, e.g., that the presence of an additional carboxylic residue in close proximity to the RSB in the rhodopsins of subgroup 2 might lead to the observed higher selectivity for sodium ions over protons and the possible absence of the proton-pumping mode. Furthermore, their kinetic studies demonstrate common features and fundamental differences between subgroup 1 and subgroup 2 of NDQ rhodopsins.

Below my major and minor concerns/points to the submitted manuscript:

Major points

- The reviewer has the impression that the chosen title: ‘A novel subgroup of light-driven sodium pumps with an additional Schiff base counterion’ might not be appropriate. Specifically, the last part: ‘an additional Schiff base counterion’. Of course, this part of the title sounds very attractive and important, but might be a bit oversold and misleading. Looking at Fig. 5C-E D105 makes a nice interaction with the RSB (2.8 Å). However, E64 (the newly identified glutamic acid) is in significant distance from RSB and cannot, in my opinion, be considered a Schiff base counterion. The description in the Abstract is much more appropriate/correct, i.e., ‘...an additional glutamic acid residue in the

close vicinity to the retinal Schiff base.'

Please indicate the distances between the oxygen atoms of E64 to the nitrogen atom of RSB (Fig. 5D and 5E). If the distances and interaction geometries to nitrogen atom of RSB are large and not optimal, the title and wording in the manuscript like 'additional RSB counterion' would have to be changed.

- Legend to Figure 2, panel B and in general, text in the following context: Are this data from ErNaR with or without EYFP? If, of the construct with EYFP, then indicate this in text (Fig. 2 legend and main text) and, most importantly, provide evidence that the fluorescent ErNaR version protein does not influence the function (e.g., pumping efficiency) or oligomeric state of the ErNaR-EYFP protein (only oligomers or also monomers because of EYFP?). This data should prove that the ErNaR-EYFP function and oligomeric state is as for wild-type: this important piece of information is currently missing. Suggestions: Functional experiments with ErNaR not fused to EYFP and fluorescence-detection size-exclusion chromatography of detergent-solubilized, purified or enriched ErNaR-EYFP protein. Other experiments leading to the same control data are of course welcome and acceptable.

- Lines 238-239, text: 'An additional 6.5 nm red-shift is observed upon further acidification to pH 2.3 (λ_{\max} 545.0 nm) (Fig. S1).' Please evaluate if this extreme low pH using detergent-solubilized, purified protein makes sense in a general context/is informative to the reader. Provide evidence that the oligomeric state is not disrupted and/or partially denatured after exposure at such extreme low pH, e.g., by using SEC. Only then, the pH at 2.3 experiments would make sense and comparable to the other pHs in the described experiment.

- Lines 253-255, text: 'On the contrary, the detergent-solubilized ErNaR remains pentameric at both pH 4.3 and 8.0 as clearly observed in size-exclusion chromatography and cryo-EM experiments.' Please provide the experimental data, e.g., elution profiles of ErNaR at pH 4.3 and 8.0 from SEC – currently not provided, missing important experimental data.

- Lines 542-543, text: 'At low pH values the pentameric assembly of KR2 is disturbed and oligomers partially dissociate into monomers in detergent micelles (Fig. 6).' Since this is an artificial situation, not reflecting the physiological environment, i.e., a microbial rhodopsin in a lipid bilayer, the reviewer suggest to minimize or even remove such statements, which would be misleading to readers.

Minor points

- Sentence in lines 93,95: Please put the Uniprot entry ID (technical information) into Material and Methods, not in the last, summarizing paragraph of «main».

- Structures of KR2 in Fig. 1: Please indicate in Figure legend PDB ID code and reference of the used model coordinates.

- Please clarify sentence in lines 122-125, i.e., '...is either absent or substituted with glutamic acid...' – currently suboptimal. From Fig. 1B I see that there are also other possible substitutions in Subgroup 2 (e.g., M, F and T instead of E), i.e., not only E or 'absent' as in BeNaR (subgroup 1).

- Lines 137-138, text: 'B. Sequence alignment of the key regions of the representative NDQ rhodopsins'. How did the authors define 'representative NDQ rhodopsins'? Please indicate in Figure legend criteria for 'representative' selection.

- Fig. 4, panel B: please indicate in schematic illustration (in blue) where the cytoplasmic and where the extracellular sides are (i.e., add corresponding text above and below schematic representation).

- Fig. 4, panels C, E and F: Please indicate distances in Å (black broken lines). This is important, e.g., to evaluate the relevance of H-bonds.

- Fig. 5: Please i) indicate where the extracellular and intracellular sides are in panel A and ii) adapt the 2Fo-Fc electron density maps (black mesh), currently too dark colour in panel C. For example, use a brighter, less strong colour such as a gray tone (instead of black).

- Line 945, text: 'In particular, the solubilized protein (60 mg/ml) in ...': Please clearly mention in which detergent the solubilized protein is. I guess DDM, but currently not clear.

Reviewer #3 (Remarks to the Author):

Microbial rhodopsins are ubiquitous photoreceptive proteins in the microbial world, and the family is still expanding with the constant discovery of novel members. Podoliak et al. identified and analyzed a novel group of Na⁺ pumping rhodopsins (NaRs) that conserve an additional acidic residue in the reaction center. Ion-transporting rhodopsins conserve charged residues in their appropriate positions to transport specific ions in specific direction. Thus, this novel NaR was expected to have unique features. Indeed, analysis of one member named ErNaR revealed that this protein can function over a wide pH range and is highly selectivity for Na⁺. These findings are of great interest, and subsequent structural analyses have provided some of the evidences for the unique features. Thus, this study should be of interest to many researchers. I'd like to request the authors to address the following points to further assure the authors' findings.

Major points:

1) Na⁺ pumping activity at acidic pH

The most important findings is that unlike well-known NaRs such as KR2, ErNaR maintains Na⁺ pumping activity even at acidic pH. However, the experimental data in Figure 2 are less convincing because the lowest pH of the activity measurement was "5" and not so acidic. The authors referred to the corresponding data for KR2 in reference #19, which does not include data at pH 5. Those data show that KR2 loses activity at pH 4.3 but is fully active at pH 6.3. To avoid the ambiguity, it would be better to compare the activities of ErNaR and KR2 at the same acidic pH in the same measurement system.

2) Discussion of the pH dependence of the absorption spectrum and the photocycle

ErNaR exhibited a slight shift in absorption spectra with decreasing pH (Fig. 3A and Fig. S1). From this observation and QM/MM simulation (Fig. S9), the authors discussed that Asp105 in the reaction center maintains its negative charge at acidic pH, thus allowing the protein to pump Na⁺ at such a low pH. However, this argument seems too simplistic. The spectral shift was small but certainly occurred (Fig. S1). Moreover, the photocycle at pH 4.3 differed significantly from that at pH 8 (Figs. S3 and S4). These pH dependences probably reflect the protonation of a residue in the reaction center. Thus, authors should discuss the origin of the pH dependences of spectral shift and photocycle.

3) Mutation of the additional acidic residue Glu64

To confirm the impact of this residue, it would be better to replace it with the corresponding amino acid of KR2.

4) Dependence of photocycle on Na⁺ concentration at acidic pH

Figure 3G indicates that high concentration of Na⁺ is required to make clear the Na⁺-concentration dependence of the photocycle at pH 4.3. These data seem to be consistent with Na⁺ pumping ability even at low pH. However, no corresponding data for KR2 have been reported. Thus, to clarify the difference between ErNaR and KR2, the same experiments should be performed for KR2.

5) Discussion based on the transient absorption changes

O1 and O2 intermediates should have similar absorption spectra. Thus I could not understand why the authors could discuss the O1 to O2 transition and their spectral difference. The authors also discussed the retinal configurations of O1 and O2 only with reference to the absorption change at 335 nm. The authors should also explain why the retinal configuration could be discussed.

Minor points:

1) The authors should explain why they chose ErNaR as the target for the analyses of molecular details.

2) The authors should explain the following:

- Method to determine the life times of the intermediates in Fig. 3E and E
- Definition and the method to determine the "LDM" (lifetime density map or lifetime distribution maps) in page 26 and in Figs. S2-S4, and S6
- Definition of $O_{1/2}$.

Point-by-point responses.

We appreciate all Reviewer comments, which significantly improved the quality and clarity of our manuscript. We have addressed all points, performed new experiments, and thoroughly revised our manuscript. All changes in the revised manuscript are marked in yellow.

Reviewer #1 (Remarks to the Author):

In their manuscript „A novel subgroup of light-driven sodium pumps with an additional Schiff base counter ion”, Podolaki, Lamm et al. perform a comprehensive biophysical analysis of a new sodium-pumping microbial rhodopsin with an interesting Schiff base counter ion. To point this out upfront: the presented analysis is as comprehensive as one may be. The manuscript reports on patch-clamp measurements, time-resolved UV/Vis measurements, both cryoEM and crystal structures at two different pH values each and high-level QM/MM calculations. I myself am a theoretical biophysicist and therefore can only comment on the reliability of the QM/MM calculations, but I definitively support a publication in Nature Communications.

We would like to thank the Reviewer for this positive evaluation and appreciation of our work.

My comments:

-- I would like the authors to prepare a subpanel directly in Fig. 1 with a structure of the protein that sketches the assumed sodium pumping path, the involved crucial residues and names the different helices. All this information is partitioned over the remaining figures and supplementary figures, but it would be very helpful to have this information available directly at the start.

We thank the Reviewer for this suggestion. We included a new panel to Fig. 1 (Fig. 1C in the revised manuscript) showing the overall side view of the KR2 protomer with the putative sodium pathway indicated with a black arrow. The key amino acid residues are shown. The helices are indicated with black capital letters. The figure legend was modified accordingly.

-- Concerning the QM/MM calculation method section: The method description is a bit short to understand what you were actually doing. Were the QM/MM simulations all performed only in Orca? This is not completely clear to me from the description in the methods.

This is an important point; we have improved the clarity and extended our description in the “Molecular simulations” section of Methods. All hybrid QM/MM energy minimizations were performed in Orca v5.0.3. The QM/MM calculations of vertical excitation energies were performed in Turbomole v7.3. The revised section is shown below:

“The computational model of ErNaR is based on X-ray crystallographic structure with PDB ID 8QLE, reported in this work. The protonation states were determined using propKa online service⁶⁰. The protons were added using pdb4amber program from AmberTools package⁶¹. To remove bad contacts energy minimization using the ff14SB AMBER force field⁶² was carried out.

The protonation states of the residues E64, D105, and D242 are determined from all possible combinations of their protonation states. We refer to them as protonation patterns. There is only 1 pattern where E64, D105, and D242 are deprotonated (zero-protonation). However, in case that one carboxylic acid side chain is protonated there are four different initial orientations to place the proton (Fig. S12C). Therefore, 12 structures were considered for one protonation of these three side chains (single-protonation) and 48 structures for two protonated side chains (double-protonation). Each structure of a specific protonation pattern was energy minimized using the hybrid quantum mechanics/molecular mechanics (QM/MM) method.^{63,64}

The QM/MM geometry optimization was performed using the L-BFGS⁶⁵ algorithm. The protonated retinal Schiff base including part of the lysine sidechain is part of the QM partition. The QM-MM boundary is placed between the C ϵ -C δ position. In addition, the side chains of the residues E64, D105, D242, and a water molecule were part of the QM region. In this case, the QM-MM boundary is put at the C α -C β bond. The MM partition comprises the remaining protein. During the geometry optimization all side chains within 5 Å distance of the retinal were flexible to move, while the rest was frozen. The QM part was treated using the B3LYP⁶⁶ density functional and Resolution-of-Identity⁶⁷ approximation, with Grimme⁶⁸ dispersion correction for the geometry optimization. The correlation-consistent⁶⁹ cc-pVDZ atomic basis set and the corresponding auxiliary basis set were used. The protein was treated using the ff14SB AMBER force field⁷⁰. The QM and MM partitions interact through electrostatic embedding. All energy minimizations were performed with the Orca v5.0.3⁷¹ computational package.

The vertical excitation energies were computed for each optimized protonation pattern using the RI second-order Algebraic Diagrammatic Construction scheme (RI-ADC(2)) for the QM part. The RI-ADC(2) calculations of vertical excitation energies were performed with Turbomole7.3 computational package⁷². Protein preparation was done using AMBER tools⁶².”

What starting structures did you use, cryoEM as well as crystal structures? Did you try different input structures? If yes, did they converge?

The X-ray crystal structure (PDB ID 8QLE) was used as a starting point for all QM/MM calculations. The reason for this choice is the significantly higher resolution of the X-ray crystal structure (1.7 Å) compared to the cryo-EM structure in our work

(2.5 Å). We have added this information at the beginning of the “Molecular simulations” section of Methods as follows:

“The computational model of ErNaR is based on X-ray crystallographic structure with PDB ID 8QLE, reported in this work.”

What was the exact simulation approach, did you perform QM/MM dynamics or only structural minimizations?

After the initial preparation of the X-ray crystal structure for the simulation, an MM energy minimization was performed. After that, a hybrid QM/MM energy minimization was done. We did not perform molecular dynamic simulations. Please note that this information has been added to the revised “Molecular simulations” section.

-- Fig. S9: If the “GS energy” in subpanel A is a potential energy, then model 6 has the highest, not the lowest energy. Did you maybe multiply the numbers with an erroneous factor of -1 ?

We thank the Reviewer for noticing this error of the sign in the relative energies. We corrected the error in the revised version of Fig. S12A.

Furthermore, the different protonation states require the addition of one or two protons into your box. That means you have additional (beneficial) proton-electron interactions in the Hamiltonian that lower the potential energy of the system. Isn't the observation that the “-1” states have the lowest GS energy compared to “-2” and “-3” then trivial, because you deal here with an open system instead of a closed one?

The Reviewer is right about the comparison of the energies between different protonation models. Since the three carboxylic acids are part of the QM region it is not possible to compare the energies with different numbers of atoms and charges. Therefore, we can only compare the relative energies for the same number of protons or protonated sidechains. In the revised Fig. S12A, we have highlighted different total charges in different colors.

Considering the accuracy of the ADC(2) method of about 0.2-0.3 eV in the comparison of the vertical excitation energies, we can conclude that the zero and one protonation models result in excitation energies which are too high compared to the experiment. For details, please also see our response to your next question below.

How robust are your estimates of the absorption maximum in nm, and how large is the error bar in comparison to experiment based on the method and level of theory you use?

We thank the Reviewer for this important question. The ADC(2) method has been benchmarked for a wide range of organic molecules similar to retinal protonated Schiff base. A benchmark from the Dreuw group [1] reported a mean absolute error of 0.29 eV for singlet vertical excitation energies compared to the best theoretical estimates. The Haettig group [2] reported a mean absolute error of 0.08 eV for adiabatic excitation energies compared to experimental values. Given these benchmarks, we conclude that the double protonated model results in excitation energies significantly closer to the experiments, which falls within the range of error of the ADC(2) method. Within the doubly protonated models, pattern 6 has the lowest ground state QM/MM energy.

References:

1. Harbach, Philipp HP, Michael Wormit, and Andreas Dreuw. "The third-order algebraic diagrammatic construction method (ADC (3)) for the polarization propagator for closed-shell molecules: Efficient implementation and benchmarking." *The Journal of Chemical Physics* 141.6 (2014).
2. Winter, Nina OC, et al. "Benchmarks for 0–0 transitions of aromatic organic molecules: DFT/B3LYP, ADC (2), CC2, SOS-CC2 and SCS-CC2 compared to high-resolution gas-phase data." *Physical Chemistry Chemical Physics* 15.18 (2013): 6623-6630.

Reviewer #2 (Remarks to the Author):

Podoliak et al. identified a new subgroup (subgroup 2) of light-driven sodium pumps (NDQ rhodopsins) bearing in addition to the two known aspartic acid residues a glutamic acid in close vicinity to the retinal Schiff base (RSB). The authors selected ErNaR from *Erythrobacter* sp. HL-111 as representative member of subgroup 2, and performed extensive functional and structural studies. Based on the obtained and presented data, the authors suggest, e.g., that the presence of an additional carboxylic residue in close proximity to the RSB in the rhodopsins of subgroup 2 might lead to the observed higher selectivity for sodium ions over protons and the possible absence of the proton-pumping mode. Furthermore, their kinetic studies demonstrate common features and fundamental differences between subgroup 1 and subgroup 2 of NDQ rhodopsins.

Below my major and minor concerns/points to the submitted manuscript:

Major points

- The reviewer has the impression that the chosen title: 'A novel subgroup of light-driven sodium pumps with an additional Schiff base counterion' might not be appropriate. Specifically, the last part: 'an additional Schiff base counterion'. Of course, this part of the title sounds very attractive and important, but might be a bit oversold and misleading. Looking at Fig. 5C-E D105 makes a nice interaction with the RSB (2.8 Å). However, E64 (the newly identified glutamic acid) is in significant distance from RSB and cannot, in my opinion, be considered a Schiff base counterion. The description in the Abstract is much more appropriate/correct, i.e., '...an additional glutamic acid residue in the close vicinity to the retinal Schiff base.'

Please indicate the distances between the oxygen atoms of E64 to the nitrogen atom of RSB (Fig. 5D and 5E). If the distances and interaction geometries to nitrogen atom of RSB are large and not optimal, the title and wording in the manuscript like 'additional RSB counterion' would have to be changed.

We thank the Reviewer for this comment. We agree that in the original version of the manuscript the argumentation for the E64 naming as the RSB counterion was missing. To fix that, first, we indicated the E64-RSB distances in Fig. 5D,E. The distances are similar at both pH 4.6 and 8.8 and are 3.7 Å. For instance, in case of a classical microbial rhodopsin BR, the distance between the RSB and D85 is 3.8 Å and between the RSB and D212 is 3.7 Å [1]. There are also not direct but water-mediated interactions of the RSB with either D85 or D212. Nevertheless, both D85 and D212 are considered as the RSB counterions. Following this logic, the RSB-E64 distance allows us to name the E64 residue as one of the RSB counterions. Second, our newly performed mutational analysis presented in the revised version of the manuscript shows that the substitution of E64 by leucine and glutamine alters the absorption maximum of the rhodopsin. Moreover, mutation of E64 results in notable pH sensitivity of the absorption spectra of the ErNaR variants, showing a direct effect of the E64 on the RSB

and retinal. Taken together, our data presented in the revised manuscript further supports our claim that E64 acts as one of the RSB counterions.

To improve the clarity of the E64's role to the broad readership of Nature Communications, we added a new section "Role of the E64 residue in the active center of ErNaR" to the revised version of the manuscript. In this section, we discuss the role of E64 in the lowering of the pKa value of D105 and show the supporting experimental data. Specifically, the explanation of the E64 role as one of the RSB counterions is discussed in the last paragraph as follows:

"Lastly, the RSB-E64 distance of only 3.7 Å (Fig. 5D,E), together with the direct effects of the E64Q and E64L mutations on the absorption spectrum (Fig. S13) and its pH sensitivity of the protein, validate our assignment of E64 as an additional RSB counterion in ErNaR. Since the E64 is conserved within the Subgroup 2 of NDQ rhodopsins we also suggest the same role of the glutamic acid in other members of this subgroup."

- Legend to Figure 2, panel B and in general, text in the following context: Are this data from ErNaR with or without EYFP? If, of the construct with EYFP, then indicate this in text (Fig. 2 legend and main text) and, most importantly, provide evidence that the fluorescent ErNaR version protein does not influence the function(e.g., pumping efficiency) or oligomeric state of the ErNaR-EYFP protein (only oligomers or also monomers because of EYFP?). This data should prove that the ErNaR-EYFP function and oligomeric state is as for wild-type: this important piece of information is currently missing. Suggestions: Functional experiments with ErNaR not fused to EYFP and fluorescence-detection size-exclusion chromatography of detergent-solubilized, purified or enriched ErNaR-EYFP protein. Other experiments leading to the same control data are of course welcome and acceptable.

We thank the Reviewer for this important suggestion. Indeed, originally, we expressed ErNaR in the genetic construct containing EYFP. The full construct was described in the beginning of the corresponding section of Results and Discussion and also in Methods as follows: C2C1-ErNaR-TS-EYFP-ES. The Reviewer's comment made us realize that our usage of the short abbreviation "ErNaR" in this section might have been misleading as it did not represent the full construct. Therefore, we have renamed the C2C1-ErNaR-TS-EYFP-ES construct as "enhanced ErNaR", or "eErNaR" by analogy with the previously reported enhanced version of KR2 (eKR2) [2]. We updated the text accordingly.

Furthermore, as suggested by the Reviewer, we have conducted functional experiments to validate that ErNaR is still functional without fused EYFP. Our findings showed that ErNaR indeed exhibits similar functional properties without EYFP. However, its targeting to the plasma membrane without EYFP was significantly poorer, which led to decrease in the photocurrent sizes. Please note that we also had to remove endoplasmic

reticulum export signal (ES), which was downstream of EYFP, likely accounting for the reduced membrane targeting.

We also thank the Reviewer for the comment on the oligomerization state of ErNaR fused with EYFP. The identification of the oligomeric state of microbial rhodopsins in the cell membrane is a general challenge and most often requires using non-natural systems. Only very recently the oligomeric state of the most-used optogenetic tool, channelrhodopsin-2 (ChR2) was studied directly in the cell membrane using quantitative photoactivated localization microscopy [3]. Thus, answering the question on the oligomeric state of ErNaR and ErNaR fused with EYFP in the plasma membrane, which would be the only directly relevant system to our electrophysiology studies, is indeed beyond the scope of our work. At the same time, our newly added data on the functionality of ErNaR without fused EYFP to that of the rhodopsin fused with EYFP shows that even in the case of EYFP effect on the oligomeric state it does not influence the ion transport function of the protein. Nevertheless, to avoid possible confusion of the Reader we have toned down the statements on the functional relevance of the pentameric form of ErNaR and KR2.

We have provided the above-discussed results of this additional study in Fig. S1 (see below) and a separate paragraph in the main text as follows:

“To verify that ErNaR can function in mammalian cells without fused fluorescent protein EYFP, we conducted an additional experiment. We transfected HEK293T cells with C2C1-ErNaR-TS-FLAG mRNA and monitored the localization of the expressed protein by staining the cells with antibody against FLAG-tag. Subcellular expression of C2C1-ErNaR-TS-FLAG was mostly restricted to intracellular compartments (Fig. S1A), presumably, due to the lack of endoplasmic reticulum export signal (ES). To label the cells that express ErNaR for electrophysiological studies, we cotransfected the cells with two separate mRNAs - C2C1-ErNaR-TS-FLAG and EYFP. We kept extracellular conditions identical to our previous experiments (pH_e 7.5 and 110 mM $[Na^+]_e$) and measured the photocurrents at pH_i 7.5 and two intracellular sodium concentrations (130 and 0 mM $[Na^+]_i$). In the presence of intracellular sodium (130 mM $[Na^+]_i$) the photocurrent showed nonlinear voltage dependence, similar to what we report for eErNaR (Fig. S1C). However, the amplitude of photocurrent decreased compared to eErNaR, likely due to the poor membrane targeting of C2C1-ErNaR-TS-FLAG. In the absence of intracellular sodium (0 mM $[Na^+]_i$) the photocurrents of C2C1-ErNaR-TS-FLAG diminished to near-zero values. The observed statistically significant ($P=0.0006$) difference in photocurrent amplitudes between 130 mM and 0 mM $[Na^+]_i$ at +60 mV (Fig. S1D) confirms our findings from experiments with eErNaR.”

Fig. S1. Functional characterization of C2C1-ErNaR-TS-FLAG in HEK293T cells. **A.** Representative confocal images of HEK293T cells expressing C2C1-ErNaR-TS-FLAG. The AlexaFluor488-conjugated FLAG-tag antibody is shown in green; nucleus stain Hoechst 33342 - in magenta. Scale bars, 10 μm. **B.** Representative photocurrents of C2C1-ErNaR-TS-FLAG recorded from HEK293T cells at 130 mM (top) and 0 mM (bottom) intracellular [Na⁺]_i and pH_i 7.5. **C.** Voltage dependence of the stationary photocurrents of C2C1-ErNaR-TS-FLAG at intracellular [Na⁺]_i 130 mM (magenta) and 0 mM (gray) and pH_i 7.5 (LJP-corrected; normalized to respective cell capacitance; mean ± SEM of n = 6-7 cells). **D.** Stationary photocurrents of C2C1-ErNaR-TS-FLAG at +60 mV, normalized to respective cell capacitance (mean ± SEM and individual data points of n = 6-7 cells). Data were extracted from the recordings at different intracellular [Na⁺]_i described in **C**. Normalized currents were analyzed using Mann-Whitney test (***) *P* < 0.001). All patch-clamp experiments were conducted at 110 mM extracellular [Na⁺]_e, pH_e 7.5; LED light with maximum at 550 nm was applied for 1 s at 34.3 mW/mm² irradiance.

- Lines 238-239, text: ‘An additional 6.5 nm red-shift is observed upon further acidification to pH 2.3 (λ_{max} 545.0 nm) (Fig. S1).’ Please evaluate if this extreme low pH using detergent-solubilized, purified protein makes sense in a general context/is informative to the reader. Provide evidence that the oligomeric state is not disrupted and/or partially denatured after exposure at such extreme low pH, e.g., by using SEC. Only then, the pH at 2.3 experiments would make sense and comparable to the other pHs in the described experiment.

We thank the Reviewer for pointing out this important question. We showed the SEC profiles of ErNaR at pH 2.3 and 8.0 in Fig. S3 of the revised manuscript. The SEC profile showed that the pentameric ErNaR remains intact even at this extremely low pH.

- Lines 253-255, text: ‘On the contrary, the detergent-solubilized ErNaR remains pentameric at both pH 4.3 and 8.0 as clearly observed in size-exclusion chromatography and cryo-EM experiments.’ Please provide the experimental data, e.g., elution profiles of ErNaR at pH 4.3 and 8.0 from SEC – currently not provided, missing important experimental data.

In line with our reply to the previous question, and in addition to the SEC profiles shown in the Fig. S3 of the revised manuscript, we modified the text in the manuscript as follows:

”It should be noted that the detergent-solubilized ErNaR remains pentameric at both pH 2.3 and 8.0 as clearly observed in size-exclusion chromatography (Fig. S3A). Cryo-EM experiments at pH 4.3 and 8.0 also showed the rhodopsin in the pentameric form. Thus, the observed minor spectral shift is the feature of the pentameric form of ErNaR and is not connected to the change of the oligomeric state.”.

- Lines 542-543, text: ‘At low pH values the pentameric assembly of KR2 is disturbed and oligomers partially dissociate into monomers in detergent micelles (Fig. 6).’ Since this is an artificial situation, not reflecting the physiological environment, i.e., a microbial rhodopsin in a lipid bilayer, the reviewer suggest to minimize or even remove such statements, which would be misleading to readers.

We thank the Reviewer for this comment. Indeed, we agree that the monomeric form of both KR2 and ErNaR is artificial and only occurs under harsh conditions. Therefore, we modified the corresponding text and removed the mentioned statements from the revised version of the manuscript as follows:

“As shown for KR2, the N-out conformation is provided by the pentameric assembly¹². The N112 side chain is H-bonded to the residues of the nearby rhodopsin molecule when flipped outside of the protomer. In pentamers of KR2 at low pH, only the N-in conformation was found¹². Furthermore, in the monomer, only the N-in conformation was observed at all pH values¹² (Fig. 6). In contrast to KR2, ErNaR adapts the N-out conformation in the pH range of 4.3-8.8 in both monomeric and pentameric forms (Fig. 6). Thus, we suggest that the conformations of the residues comprising the characteristic NDQ motif (N101, D105, Q112 in ErNaR and N112, D116, Q123 in KR2) are similar in the ground states of both proteins, which is likely a common feature of the members of both Subgroups 1 and 2. At the same time, this N-out conformation seems to be more stable in Subgroup 2 than in Subgroup 1, and is independent of pH and oligomeric state.”.

Minor points

- Sentence in lines 93,95: Please put the Uniprot entry ID (technical information) into Material and Methods, not in the last, summarizing paragraph of «main».

This change is included in the revised manuscript.

- Structures of KR2 in Fig. 1: Please indicate in Figure legend PDB ID code and reference of the used model coordinates.

This change is included in the revised manuscript.

- Please clarify sentence in lines 122-125, i.e., ‘...is either absent or substituted with glutamic acid...’ – currently suboptimal. From Fig. 1B I see that there are also other possible substitutions in Subgroup 2 (e.g., M, F and T instead of E), i.e., not only E or ‘absent’ as in BeNaR (subgroup 1).

We thank the Reviewer for this comment. We have modified the sentence as follows: “For instance, the D102 residue, forming the interprotomeric sodium binding site on the extracellular surface of the KR2 pentamer, is either absent or substituted with other amino acid residues in the Subgroup 2 (Fig. 1B,D).”.

- Lines 137-138, text: ‘B. Sequence alignment of the key regions of the representative NDQ rhodopsins’. How did the authors define ‘representative NDQ rhodopsins’? Please indicate in Figure legend criteria for ‘representative’ selection.

We added a description of our reasoning in selection of the representative NDQ rhodopsins as follows:

“For the representative NDQ rhodopsins we selected biophysically characterized proteins also described in²².”.

- Fig. 4, panel B: please indicate in schematic illustration (in blue) where the cytoplasmic and where the extracellular sides are (i.e., add corresponding text above and below schematic representation).

We modified Fig. 4B according to the Reviewer’s comment.

- Fig. 4, panels C, E and F: Please indicate distances in Å (black broken lines). This is important, e.g., to evaluate the relevance of H-bonds.

We modified Fig. 4C,E,F following the Reviewer’s comment.

- Fig. 5: Please i) indicate where the extracellular and intracellular sides are in panel A and ii) adapt the 2Fo-Fc electron density maps (black mesh), currently too dark colour in panel C. For example, use a brighter, less strong colour such as a gray tone (instead of black).

We modified Fig. 5 following the Reviewer's comment.

- Line 945, text: 'In particular, the solubilized protein (60 mg/ml) in ...': Please clearly mention in which detergent the solubilized protein is. I guess DDM, but currently not clear.

We have corrected the protein concentration and modified the sentence following the Reviewer's comment as follows:

"In particular, the protein at 70 mg/ml solubilized in DDM in the crystallization buffer was mixed with premelted at 42°C monoolein (MO, Nu-Chek Prep) in a 3:2 ratio (lipid:protein) to form a lipidic mesophase."

References:

1. Borshchevskiy, V., Kovalev, K., Round, E. *et al.* "True-atomic-resolution insights into the structure and functional role of linear chains and low-barrier hydrogen bonds in proteins." *Nat Struct Mol Biol* **29**, 440–450 (2022).
2. Grimm, C., Silapetere, A., Vogt, A. *et al.* „Electrical properties, substrate specificity and optogenetic potential of the engineered light-driven sodium pump eKR2." *Sci Rep* **8**, 9316 (2018).
3. Gensch, T., Bestsennaia, E., Maslov, I., *et al.* „Revealing the Oligomerization of Channelrhodopsin-2 in the Cell Membrane using Photo-Activated Localization Microscopy." *Angewandte Chemie* **2024** e202307555.

Reviewer #3 (Remarks to the Author):

Microbial rhodopsins are ubiquitous photoreceptive proteins in the microbial world, and the family is still expanding with the constant discovery of novel members. Podoliak et al. identified and analyzed a novel group of Na⁺ pumping rhodopsins (NaRs) that conserve an additional acidic residue in the reaction center. Ion-transporting rhodopsins conserve charged residues in their appropriate positions to transport specific ions in specific direction. Thus, this novel NaR was expected to have unique features. Indeed, analysis of one member named ErNaR revealed that this protein can function over a wide pH range and is highly selectivity for Na⁺. These findings are of great interest, and subsequent structural analyses have provided some of the evidences for the unique features. Thus, this study should be of interest to many researchers. I'd like to request the authors to address the following points to further assure the authors' findings.

We thank the Reviewer for the high evaluation of our work. The raised points are addressed below.

Major points:

1) Na⁺ pumping activity at acidic pH

The most important findings is that unlike well-known NaRs such as KR2, ErNaR maintains Na⁺ pumping activity even at acidic pH. However, the experimental data in Figure 2 are less convincing because the lowest pH of the activity measurement was "5" and not so acidic. The authors referred to the corresponding data for KR2 in reference #19, which does not include data at pH 5. Those data show that KR2 loses activity at pH 4.3 but is fully active at pH 6.3. To avoid the ambiguity, it would be better to compare the activities of ErNaR and KR2 at the same acidic pH in the same measurement system.

We thank the Reviewer for this comment. We should clarify that our work is primarily a thorough biophysical and structural characterization of ErNaR rather than a comparative study of NDQ rhodopsins. In addition to that, our data on ErNaR and available literature data on KR2 allowed us to compare these representatives of the two subgroups of NDQ rhodopsins in some aspects. Clearly, further studies are required to make a complete comparison of these subgroups and our work serves as a basis for such investigations. We added this statement to the revised version of the manuscript as shown at the end of this reply.

We also agree that one of the key findings of our work is the ability of ErNaR to transport sodium even at pH as low as 5.0. We would also like to state that pH 5.0 is rather acidic for HEK cells in which the electrophysiology experiments were performed, as well as for other mammalian cells. The ability of ErNaR to pump sodium at pH 5.0 is the first demonstration of the NDQ rhodopsin's functioning under such harsh conditions, which was never demonstrated for KR2 or any other member of Subgroups 1 and 2. Therefore, we agree with the Reviewer that available literature

data on KR2 is not sufficient for direct comparison of its functional properties to those of ErNaR.

Nevertheless, in line with Point 4 raised by the Reviewer, we studied the KR2 photocycle sensitivity to the sodium concentrations at pH 4.3, which clearly showed principally different behavior of KR2 and ErNaR at acidic pH values (more details are given in the reply to Point 4). This additionally supports our hypothesis on the different functionalities of KR2 and ErNaR at low pH.

Taken together, we accounted for the newly obtained data on KR2 and toned down the statements in the “Comparison of the two subgroups of NaRs” section of the revised manuscript as follows:

“Electrophysiology demonstrated that ErNaR is capable of active transport of sodium across the membrane in a wide range of pH values, including acidic pH as low as 5, which has not been shown for KR2 or other members of Subgroup 1 of NDQ rhodopsins. Thus, the functional data available on KR2 in literature does not allow the direct comparison of its properties to ErNaR. Nevertheless, in KR2, as well as in other members of Subgroup 1, the pH decrease leads to the protonation of the main RSB counterion, aspartic acid of the characteristic NDQ motif^{1,3}. This protonation correlates with the deceleration of the photocycle and loss of the O-states reflecting the lowered sodium binding efficiency^{1,32}. On the contrary, ErNaR lacks such pH dependence on absorption spectra and retains the sodium-dependent O-states even at pH 4.3. Therefore, the cumulative data allow us to suggest that the presence of an additional carboxylic residue in close proximity to the RSB in the rhodopsins of Subgroup 2 might contribute to their efficient functioning as sodium pumps in a wider range of pH values than that of the Subgroup 1 members. Further comparative studies of the NDQ rhodopsins from both subgroups are required to validate this hypothesis.”.

2) Discussion of the pH dependence of the absorption spectrum and the photocycle

ErNaR exhibited a slight shift in absorption spectra with decreasing pH (Fig. 3A and Fig. S1). From this observation and QM/MM simulation (Fig. S9), the authors discussed that Asp105 in the reaction center maintains its negative charge at acidic pH, thus allowing the protein to pump Na⁺ at such a low pH. However, this argument seems too simplistic. The spectral shift was small but certainly occurred (Fig. S1). Moreover, the photocycle at pH 4.3 differed significantly from that at pH 8 (Figs. S3 and S4). These pH dependences probably reflect the protonation of a residue in the reaction center. Thus, authors should discuss the origin of the pH dependences of spectral shift and photocycle.

We thank the Reviewer for this comment and agree that the discussion on the pH dependence of the absorption spectrum and photocycle of ErNaR was missing in the original manuscript. We have modified the text in the “Spectroscopy of ErNaR” section of the revised manuscript and added the Discussion on these effects as follows:

“Taken together, while at pH 8.0 the behavior of ErNaR is similar to KR2 and other studied NaRs, at pH 4.3 it demonstrates unique spectral features. First, the spectral shift upon the pH decrease in ErNaR is small, being <10 nm between pH 8.0 and 2.3 and only 3 nm between pH 8.0 and 4.3. We suggest that these small spectral shifts of ErNaR are not connected to the protonation of the main RSB counterion, D105, since its protonation is expected to cause much larger spectral changes^{1,34,35}. Indeed, in KR2, the protonation of D116 upon pH decrease from 8.0 to 4.3 results in the red-shift of ~25 nm¹. In addition, the ultrafast kinetics of ErNaR is pH-independent (Fig. S4), arguing for the absence of the D105 protonation at acidic pH. Thus, we speculate that the minor red-shift of the ErNaR spectrum at low pH is associated with the protonation of rechargeable residues distant from the RSB. It could be also connected to the partial redistribution of the charges at the RSB counterion complex including residues D105, E64, and D242 as described further in the manuscript.

The second feature of ErNaR at low pH is the presence of the O-states in the photocycle as well as their sensitivity to sodium. We suggest that the preservation of the O-states supports our above-mentioned hypothesis on the deprotonated form of the RSB counterion D105 even at pH 4.3. The sensitivity of the O-states to sodium ions is also in line with the observed sodium-transport activity of ErNaR at low pH (Fig. 1C,D). We speculate that the blue-shift of the O-states at pH 4.3 compared to pH 8.0 has several reasons. Specifically, at low sodium concentrations (0 mM and 100 mM) the O-states likely reflect the binding of a proton in the active center of ErNaR due to much higher concentration of protons than that of sodium. This hypothesis is in line with the model of competitive uptake of protons and sodium ions shown for KR2³². However, as the proton-pumping activity of ErNaR was not detected at low pH, we suggest that after the binding the proton is released back to the cytoplasm in the same manner it is proposed for another sodium pump GLR³. Thus, at low sodium concentrations, the spectral and kinetic differences between the O-states of the ErNaR photocycles at pH 4.3 and 8.0 might originate from the binding of different substrates. We speculate that at high sodium concentration (1000 mM) the sodium ion is bound in the active center of ErNaR in the O-states in the same manner at pH 4.3 and 8.0. In this case, the slight blue-shift of the O-states at pH 4.3 might be caused by the protonation of the residues distant from the RSB upon acidification.”

We also added the discussion on the possible charge redistribution in the RSB counterion complex of ErNaR upon pH decrease to the “Role of the E64 residue in the active center of ErNaR” section of the revised manuscript as follows:

“In this case, only a minor spectral red-shift of ErNaR upon pH decrease (Fig. S2) might be connected to the charge redistribution with the counterion complex including the D105-E64 pair reflected in different lengths of proposed LBHB at pH 4.6 and 8.8 (2.1 and 2.2 Å, respectively) (Fig. 5D,E).”

3) Mutation of the additional acidic residue Glu64

To confirm the impact of this residue, it would be better to replace it with the corresponding amino acid of KR2.

We would like to thank the Reviewer for this very helpful comment. We agree that the mutational analysis of ErNaR at the position of E64 is beneficial to further support the proposed role of this amino acid. Therefore, we replaced E64 with the corresponding residue in KR2, resulting in the E64L mutant of ErNaR, and produced the E64L mutant for the spectroscopy studies. We measured pH-dependent absorption spectra in the pH range of 3.0 - 11.5. The results are shown in Fig. S13A. In contrast to the WT protein, but similar to KR2, we observed strong pH dependency of the absorption maximum in the mutant. Specifically, at low pH, the absorption maximum is at ~575 nm and shifts significantly upon pH increase to ~522 nm at alkaline conditions. The observed notable sensitivity of the absorption maximum to the pH value supports our hypothesis on the key role of E64 in the stabilization of the D105 residue in the WT rhodopsin. Interestingly, the spectral transition in the KR2-like E64L mutant of ErNaR occurs at higher pH than that in KR2. For instance, even at pH 8.0, the absorption maximum of ErNaR-E64L is at 568 nm compared to 525 nm of KR2. Thus, the substitution of leucine with glutamate in the Subgroup 2 members is likely not the only factor determining the spectral differences between the two subgroups of NDQ rhodopsins.

In addition, we produced the E64Q variant and studied its pH-dependent absorption spectra to validate our hypothesis on the protonation state of E64 in ErNaR. The results are shown in Fig. S13B. The position of the measured absorption maxima of the E64Q mutant is similar to that of the WT protein at acidic conditions, where the variation between both proteins is in the range of a few nm. At neutral and alkaline pH, the absorption maximum is blue-shifted in both the mutant and WT, but for the mutant the observed blue-shift is more pronounced, resulting in an absorption maximum at ~525 nm at pH > 8.0, compared to the absorption maximum at ~535.5 nm for the WT protein in the same pH range. Over the whole spectral range, an absorption shift of ~20 nm was observed for ErNaR-E64Q, compared to the 10 nm observed for the WT protein. Since the E64Q substitution mimics the rhodopsin with the protonated E64 residue, the overall similar behavior and absorption maximum positions in WT and E64Q support our hypothesis of the protonated (neutral) E64 residue in ErNaR in a wide pH range. At the same time, the increased pH sensitivity of the E64Q mutant is in line with the proposed low-barrier hydrogen bond between E64 and D105, which was suggested to stabilize the D105 residue but cannot be formed in the mutant.

In order to account for the mentioned points, we added Fig. S13 showing the pH-dependent absorption spectra of the E64L and E64Q mutants of ErNaR to the revised manuscript. We also added a new section "Role of the E64 residue in the active center of ErNaR" where we discuss possible role(s) of the E64 residue and provide experimental evidence of our hypotheses. The results on the studies of the ErNaR mutants are described as follows:

“To probe this hypothesis, we performed a mutational analysis of ErNaR. We substituted E64 with Leu and Gln and studied spectroscopic properties of the E64L and E64Q variants of the rhodopsin. The KR2-like E64L mutant showed a notable red shift of 50 nm upon pH decrease from 11 to 6, supporting our hypothesis on the key role of E64 for the spectral stability of ErNaR (Fig. S13A). Surprisingly, the shift is ~2 times larger than that of KR2 (~25 nm) and also occurs at higher pH value. This indicates that E64 is not the only determinant of the spectral differences between ErNaR and KR2. We speculate that the above-described differences in other regions of the two proteins also contribute to their spectral differences.

The E64Q mutant should mimic the ErNaR with protonated E64 and thus is expected to show similar spectral behavior to that of the wild type (WT) protein. However, it demonstrated a larger spectral shift upon pH titration (~20 nm vs. ~10nm in E64Q and WT, respectively) (Fig. S13B). The maximum absorption wavelength at neutral and alkaline pH values is also slightly different in the mutant (~525 nm) and WT (~535 nm) (Fig. S13B). We suggest that the spectral differences between the mutant and the WT protein originate from the absence of the LBHB between D105 and Q64 in ErNaR-E64Q since it cannot be formed between the Asp and Gln residues. This result additionally supports our hypothesis on the LBHB between D105-E64 and its role in the pKa lowering of D105 in ErNaR.”

4) Dependence of photocycle on Na⁺ concentration at acidic pH

Figure 3G indicates that high concentration of Na⁺ is required to make clear the Na⁺-concentration dependence of the photocycle at pH 4.3. These data seem to be consistent with Na⁺ pumping ability even at low pH. However, no corresponding data for KR2 have been reported. Thus, to clarify the difference between ErNaR and KR2, the same experiments should be performed for KR2.

We appreciate the feedback of the Reviewer and agree that sodium dependent measurements of the KR2 photocycle kinetics at low pH have not been reported. In order to compare the photocycles of KR2 and ErNaR at low pH, we investigated KR2 under the same conditions as ErNaR (Fig. S7). Since the focus is on sodium related effects and especially the impact of high sodium concentration on KR2, we used sodium concentrations 200 mM, 500 mM and 1000 mM and performed flash photolysis spectroscopy by measuring transients at characteristic wavelengths. The wavelengths have been chosen according to the flash photolysis measurement of KR2 at pH 5.5 reported in [1]. The acquired data shows sodium dependent effects on the photocycle at pH 4.3, but they significantly differ from those observed for ErNaR. Especially, the O intermediates, which are associated with sodium binding [2] are missing for KR2 at pH 4.3 at all studied sodium concentrations. Specifically, the sodium dependent effects observed for KR2 do not include any significant changes in the shape of the measured transients, but rather a decrease in intensity of the later signals, if the sodium concentration is increased. Thus, the newly obtained data on KR2 allow for the direct

comparison of ErNaR and KR2 and further stresses the differences between the rhodopsins' behavior at low pH values.

The additionally acquired data on KR2 is shown in Fig. S7 of the revised manuscript. Furthermore, we added the discussion on the comparison of the ErNaR and KR2 photocycles, especially the O state kinetics at low pH, to the "Spectroscopy of ErNaR" section of the revised manuscript as follows:

"However, in contrast to KR2, the O₁ and O₂ states were observed for ErNaR at pH 4.3 and are clearly sensitive to sodium concentrations (Fig. S6). Indeed, our flash photolysis data on KR2 at pH 4.3 showed no rise of the O state even at high sodium concentrations, such as 1000 mM NaCl (Fig. S7)."

5) Discussion based on the transient absorption changes

O₁ and O₂ intermediates should have similar absorption spectra. Thus I could not understand why the authors could discuss the O₁ to O₂ transition and their spectral difference. The authors also discussed the retinal configurations of O₁ and O₂ only with reference to the absorption change at 335 nm. The authors should also explain why the retinal configuration could be discussed.

We thank the Reviewer for this question. We agree that a proper description of the procedure of the identification and the distinguishing between the O₁ and O₂ states was missing in the original version of the manuscript. We have modified the "Spectroscopy of ErNaR" to clearly explain the origin of the two O-states as follows:

"Thus, at 100 mM NaCl, ErNaR shows mixed kinetics of one subpopulation likely undergoing the "sodium-pumping mode" photocycle while the other subpopulation undergoes a long-living M-intermediate before directly relaxing back to the parent state (Fig. S5B). For the sodium-pumping mode, analysis of the flash-photolysis data showed the presence of two O intermediates (O₁ and O₂) similar to those shown for KR2 and IaNaR^{14,15} (Fig. 3E, Fig. S5C; see Methods for more details on the data analysis)."

The corresponding methods of the time-resolved spectroscopy data analysis were added to the Methods of the revised manuscript as follows:

"Analysis of time-resolved spectroscopic data was performed using OPTIMUS software⁴⁴, available free-of-charge at www.optimusfit.org. The data of the ultrafast transient absorption and transient flash photolysis measurements were objected to the model-free lifetime distribution analysis (LDA) yielding the lifetime distributions of the individual photointermediate transitions, which are summarized in a lifetime distribution map (LDM). The lifetimes of the photocycle intermediate transitions have been determined at the maximum of the respective lifetime distribution in the corresponding LDM and are therefore mentioned as approximate values (Fig. S14).

To determine the LDMs visualized in the manuscript, a Tikhonov regularization with 200 regularization factors in the range of 0.01 to 5 were used together with 200 lifetimes to calculate the inverse Laplace transform of the experimental dataset $S(\tau, \lambda_{exc}, \lambda_i)$ as explained in more detail in Slavov et al.⁴⁴. The start lifetime was set to 0.03 ps for the measurements on the ultrafast timescale and 400 ns for measurements on the ns to s timescale. The end lifetime was set to 5400 ps for the ultrafast timescale and three times the value of the last experimental time point (58.5 s in the case of a 19.5 s measurement duration) for the ns to s timescale. The shown LDMs have been chosen via the L-curve criterion.”

We also added an explanation concerning the assignment of the retinal configurations in the O_1 and O_2 states at pH 8.0, which was originally missing in the manuscript, as follows:

“The O_1 -to- O_2 transition coincides with the decay of the signal at 335 nm (Fig. 3C; Fig. S5C). This near-UV band was previously termed as second bright state (SBS) and was shown to be a marker of the 13-cis retinal configuration in KR2²³ and inward proton pump NsXeR²⁴. Thus, our data suggest that the O_1 state of ErNaR contains 13-cis retinal, while O_2 has an all-trans retinal. These results are in line with the findings and interpretation of Fujisawa et al.¹⁷ for IaNaR.”

The same assignment was done for the data obtained at pH 4.3 and is discussed in the revised version of the manuscript as follows:

“Similar to pH 8.0, the SBS signal indicates a 13-cis configuration of the retinal for O_1 and an all-trans configuration for O_2 (Fig. 3F).”

We also added the explanation of the SBS abbreviation with corresponding references to the Fig. 3 legend as follows:

“The SBS abbreviation represents the “second bright state” signal at 335 nm being a marker of the 13-cis retinal configuration^{23,24}.”

Minor points:

1) The authors should explain why they chose ErNaR as the target for the analyses of molecular details.

We thank the Reviewer for this question. Our work aimed at obtaining high-resolution structures of a representative of the Subgroup 2 of NDQ rhodopsins to reveal details of their organization. As the main method for obtaining such structures we used X-ray crystallography. The ErNaR rhodopsin was chosen for studies due to the absence of elongated N and C termini and long unstructured loops in order to increase the chances for crystallization. We added this information to the revised version of the manuscript

as follows:

“ErNaR was selected as a promising target for obtaining high-resolution structural data using X-ray crystallography as it lacks unstructured termini and interhelical loops”.

2) The authors should explain the following:

- Method to determine the life times of the intermediates in Fig. 3E and E

Lifetimes presented in the photocycle schemes shown in Fig. 3E and 3H have been derived from the LDMs of the respective measurements. This has been done via determination of the temporal position of the maximum of each lifetime distribution accounting for a photocycle intermediate transition. Since the LDM results in lifetime distributions, the retrieved lifetimes have been mentioned as approximate values. Furthermore, to show the complementarity between the LDA and the more commonly used global lifetime analysis (GLA), we directly compared the results of both analysis methods exemplary for the flash photolysis measurement of ErNaR at pH 4.3, 1000 mM NaCl, originally shown in Fig. 3F and Fig. S4C of the manuscript.

Fig. S14. Comparison of lifetime distribution analysis and global lifetime analysis. Exemplary comparison of the results of lifetime distribution analysis (LDA) and global lifetime analysis (GLA) to show the complementarity of both analysis methods for time-resolved spectroscopic datasets. The result of the LDA is shown as the lifetime distribution map (LDM), while the results of the GLA are shown as decay-associated spectra (DAS). The via GLA obtained lifetimes have been highlighted as dashed lines in the LDM to show the comparability of both analysis methods.

The lifetimes of the photocycle intermediate transitions obtained via GLA have been highlighted as dashed lines in the LDM (Fig. S14, middle). In most cases, the GLA lifetime hits the maximum of the respective lifetime distribution in the LDM, showing that both methods lead to similar results (Fig. S14). In two cases, namely in the range of 0.1 - 1 ms and at around 100 ms, two GLA lifetimes were obtained to describe the same lifetime distribution in the LDM. This phenomenon usually reflects non-exponential kinetics. Due to the quasi-continuous sum of exponential functions used in the LDA in contrast to the discrete amount of exponential functions used in GLA, the LDA is able to describe such a kinetic behavior in a better way, as one lifetime

distribution. The 0.2 ms lifetime together with the associated lifetime distribution was not further considered due to its minor spectral contribution. Most-likely this is an artefact of the fitting procedure, when trying to describe signals that do not change much over a longer periode in time. This is similar compared to the “oscillating pattern” observed in the LDM at the end of the ultrafast timescale (Fig. S4), where no spectral changes occur.

In order to clarify the determination of the mentioned lifetimes, we added the following sentence to the manuscript in the Methods section “Analysis of time-resolved spectroscopic data” as follows:

“The lifetimes of the photocycle intermediate transitions have been determined at the maximum of the respective lifetime distribution in the corresponding LDM and are therefore mentioned as approximate values.”

- Definition and the method to determine the "LDM" (lifetime density map or lifetime distribution maps) in page 26 and in Figs. S2-S4, and S6

We added the details on the time-resolved spectroscopy data analysis to the Methods section “Analysis of time-resolved spectroscopic data” as follows:

“Analysis of time-resolved spectroscopic data was performed using OPTIMUS software⁴⁴, available free-of-charge at www.optimusfit.org. The data of the ultrafast transient absorption and transient flash photolysis measurements were objected to the model-free lifetime distribution analysis (LDA) yielding the lifetime distributions of the individual photointermediate transitions, which are summarized in a lifetime distribution map (LDM). The lifetimes of the photocycle intermediate transitions have been determined at the maximum of the respective lifetime distribution in the corresponding LDM and are therefore mentioned as approximate numbers (Fig. S14).

To determine the LDMs visualized in the manuscript, a Tikhonov regularization with 200 regularization factors in the range of 0.01 to 5 were used together with 200 lifetimes to calculate the inverse Laplace transform of the experimental dataset $S(\tau, \lambda_{exc}, \lambda_i)$ as explained in more detail in ⁴⁴. The start lifetime was set to 0.03 ps for the measurements on the ultrafast timescale and 0.0004 ms for measurements on the ns to s timescale. The end lifetime was set to 5400 ps for the ultrafast timescale and three times the last experimental time point for the ns to s timescale. The shown LDMs have been chosen via the L-curve criterion.”

- Definition of O_{1/2}.

We agree with the Reviewer that the label O_{1/2} used in various figures throughout the original manuscript could be misleading and might raise the question of the definition of the O_{1/2} intermediate state. As mentioned earlier, at most investigated conditions we identified two O-states, O₁ and O₂, which appeared spectrally similar. Subsequently, the exact determination of the spectral position, as well as placing two labels close to each other, while maintaining readability of both the labels and the experimental data is a challenge. Therefore, we indicated the region of both O intermediates as “O_{1/2}”, but a proper explanation of the used label was missing. To account for that, we added the following to the legend of Fig. 3C, and Figs. S5, S6, S8, and S9, in order to clarify the meaning of O_{1/2}.

“Due to spectral similarity, the region of the O₁ and O₂ intermediates is indicated as O_{1/2}.”

Furthermore, we replaced O_{1/2} in the legend of Fig. 3D and 3G with “O₁ and O₂ intermediates”.

References:

1. Asido, M., Kar, R.K., *et al.* “Transient Near-UV Absorption of the Light-Driven Sodium Pump Krokobacter eikastus Rhodopsin 2: A Spectroscopic Marker for Retinal Configuration.” *J. Phys. Chem. Lett.* **12**, 27, 6284–6291 (2021).
2. Inoue, K., Ono, H., Abe-Yoshizumi, R. *et al.* A light-driven sodium ion pump in marine bacteria. *Nat Commun* **4**, 1678 (2013).

Reviewer #1 (Remarks to the Author):

My comments have been sufficiently taken into account. I have one last request concerning my second-to-last comment: In the caption of Fig. S12, please indicate how many eV the experimentally observed 530 nm are, i.e., change the "(~530 nm)" into "(~530 nm, i.e., 2.34 eV)". Furthermore, replace the half-sentence "...as well as the lowest ground state energy compared to energies of other protonation systems." with "...as well as the lowest ground state energy compared to energies of the "-1" protonation systems." After these changes have been implemented, the manuscript is ready for publication from my side.

Reviewer #2 (Remarks to the Author):

The authors have carefully addressed my concerns in the revised version. Thank you and congratulations, a nice contribution.

Reviewer #3 (Remarks to the Author):

I thank the authors for their sincere replies and corrections. I noticed the following points, but they are not serious problems.

1) Figure 3D and its associated description on lines 289-291

I apologize that I should have left this comment in the previous review.

The authors determined the "Kaffinity" value (190 mM) from the plots in the right panel of Fig. 3D, where the maximum absorbance changes in left panel, which reflect the accumulations of O intermediate, are plotted against the Na⁺ concentration. The authors probably defined the "Kaffinity" as the Na⁺ concentration at which the populations of Na⁺ pumping and no pumping ErNaR are identical. However, the accumulation of O not only depends on the population of Na⁺ pumping ErNaR, but also on the rates of its formation and decay. Thus, please reconsider the definition and the name of "Kaffinity".

2) Line 318: "(Fig. 1F, G)" and "(Fig. 1F)" might be "(Fig. 3F, G)" and "(Fig. 3F)", respectively.

3) Lines 1211, 1219, 1236: "ultrafast" doesn't seem necessary.

Reviewer #1 (Remarks to the Author):

My comments have been sufficiently taken into account. I have one last request concerning my second-to-last comment: In the caption of Fig. S12, please indicate how many eV the experimentally observed 530 nm are, i.e., change the "(~530 nm)" into "(~530 nm, i.e., 2.34 eV)". Furthermore, replace the half-sentence "...as well as the lowest ground state energy compared to energies of other protonation systems." with "...as well as the lowest ground state energy compared to energies of the "-1" protonation systems." After these changes have been implemented, the manuscript is ready for publication from my side.

We thank the Reviewer for the high evaluation of our revised manuscript. We have modified the caption of Fig. S12 in accordance with the Reviewer's comment.

Reviewer #3 (Remarks to the Author):

I thank the authors for their sincere replies and corrections.

I noticed the following points, but they are not serious problems.

1) Figure 3D and its associated description on lines 289-291

I apologize that I should have left this comment in the previous review.

The authors determined the "Kaffinity" value (190 mM) from the plots in the right panel of Fig. 3D, where the maximum absorbance changes in left panel, which reflect the accumulations of O intermediate, are plotted against the Na⁺ concentration. The authors probably defined the "Kaffinity" as the Na⁺ concentration at which the populations of Na⁺ pumping and no pumping ErNaR are identical. However, the accumulation of O not only depends on the population of Na⁺ pumping ErNaR, but also on the rates of its formation and decay. Thus, please reconsider the definition and the name of "Kaffinity".

We agree with the Reviewer that the naming of the measured value as "Kaffinity" was inaccurate. We modified the name to "KO_(1/2)-saturation" to better reflect the meaning of the measured value. We modified the name in Fig. 3D and its legend.

2) Line 318: "(Fig. 1F, G)" and "(Fig. 1F)" might be "(Fig. 3F, G)" and "(Fig. 3F)", respectively.

We thank the Reviewer for noticing of this misprint. We corrected it in the revised manuscript.

3) Lines 1211, 1219, 1236: "ultrafast" doesn't seem necessary.

We thank the Reviewer for this comment. We removed the word "ultrafast" from the revised version of the manuscript.